# White Adipocyte Stem Cell Expansion Through Infant Formula Feeding: New Insights into Epigenetic Programming Explaining the Early Protein Hypothesis of Obesity

**DOI:** 10.3390/ijms26104493

**Published:** 2025-05-08

**Authors:** Bodo C. Melnik, Ralf Weiskirchen, Swen Malte John, Wolfgang Stremmel, Claus Leitzmann, Sabine Weiskirchen, Gerd Schmitz

**Affiliations:** 1Department of Dermatology, Environmental Medicine and Health Theory, University of Osnabrück, D-49076 Osnabrück, Germany; sjohn@uos.de; 2Institute of Molecular Pathobiochemistry, Experimental Gene Therapy and Clinical Chemistry (IFMPEGKC), RWTH University Hospital Aachen, D-52074 Aachen, Germany; sweiskirchen@ukaachen.de; 3Institute for Interdisciplinary Dermatological Prevention and Rehabilitation (iDerm), University of Osnabrück, D-49076 Osnabrück, Germany; 4Praxis for Internal Medicine, D-76530 Baden-Baden, Germany; wolfgangstremmel@aol.com; 5Institut für Ernährungswissenschaft, Universität Gießen, D-35392 Gießen, Germany; claus@leitzmann-giessen.de; 6Institute of Clinical Chemistry and Laboratory Medicine, University Hospital of Regensburg, D-93053 Regensburg, Germany; gerd.schmitz@ukr.de

**Keywords:** adipocyte stem cell, adipogenesis, breastfeeding, enhancer of zeste homolog 2, epigenetic regulation, formula feeding, fat mass- and obesity-associated gene, mechanistic target of rapamycin complex 1, obesity, S6 kinase 1, wingless signaling

## Abstract

Prolonged breastfeeding (BF), as opposed to artificial infant formula feeding (FF), has been shown to prevent the development of obesity later in life. The aim of our narrative review is to investigate the missing molecular link between postnatal protein overfeeding—often referred to as the “early protein hypothesis”—and the subsequent transcriptional and epigenetic changes that accelerate the expansion of adipocyte stem cells (ASCs) in the adipose vascular niche during postnatal white adipose tissue (WAT) development. To achieve this, we conducted a search on the Web of Science, Google Scholar, and PubMed databases from 2000 to 2025 and reviewed 750 papers. Our findings revealed that the overactivation of mechanistic target of rapamycin complex 1 (mTORC1) and S6 kinase 1 (S6K1), which inhibits wingless (Wnt) signaling due to protein overfeeding, serves as the primary pathway promoting ASC commitment and increasing preadipocyte numbers. Moreover, excessive protein intake, combined with the upregulation of the fat mass and obesity-associated gene (*FTO*) and a deficiency of breast milk-derived microRNAs from lactation, disrupts the proper regulation of *FTO* and Wnt pathway components. This disruption enhances ASC expansion in WAT while inhibiting brown adipose tissue development. While BF has been shown to have protective effects against obesity, the postnatal transcriptional and epigenetic changes induced by excessive protein intake from FF may predispose infants to early and excessive ASC commitment in WAT, thereby increasing the risk of obesity later in life.

## 1. Introduction

Despite numerous research and public policy efforts, the obesity epidemic continues to threaten human health. Childhood obesity has already reached a pandemic [1,2]. Overweight and obesity in adults are linked to chronic diseases such as type 2 diabetes mellitus (T2DM), heart disease, stroke, hypertension, and certain cancers, posing a significant public health burden [3,4]. In 2015, a total of 107.7 million children and 603.7 million adults were obese. Since 1980, the prevalence of obesity has doubled in over 70 countries and has steadily increased in most others [4]. According to WHO in 2022, 37 million children under the age of 5 years were overweight and over 390 million children and adolescents aged 5–19 years were overweight, with 160 million living with obesity [5]. By 2025, global obesity rates are predicted to reach 18% in men and exceed 21% in women [6,7].

The postnatal period, during which human infants are physiologically exposed to human breast milk (HBM), is a critical window for determining long-term metabolic health [8]. In 2005, Koletzko and his coworkers [9] hypothesized that high early protein intakes exceeding metabolic requirements enhance weight gain in infancy and increase the risk of later obesity, known as the “early protein hypothesis”. However, the molecular pathways explaining the link between excessive postnatal protein overfeeding and obesity development have not yet been clearly delineated. The aim of this review is to explore the missing molecular link between postnatal protein overfeeding and the subsequent transcriptional and epigenetic changes that accelerate adipocyte stem cell (ASC) expansion in the adipose vascular niche during postnatal white adipose tissue (WAT) development.

It has been recognized that nutrition in early life can alter the epigenome [10], leading to different phenotypes and altering disease susceptibilities [11]. The quantity and quality of nutrition during neonatal life plays a critical role beyond prenatal development programming for long-term health or disease [8]. HBM plays a crucial role in epigenetic programming [12]. Experimental evidence in mice suggests that factors altering epigenetic mechanisms can be transmitted through milk and passed down to future generations [13]. Bioinformatic analysis has demonstrated that differentially expressed microRNAs (miRs) were associated with phosphatidylinositol-3 kinase (PI3K)-AKT, ERB-B receptor tyrosine kinase (ERBB), the mechanistic target of rapamycin (mTOR), mitogen-activated protein kinase 1 (MAPK), and insulin signaling pathways [13].

Environmental factors during prenatal and postnatal development have the potential to impact the structure and function of adipose tissue (AT), which can influence the development of obesity [14], and, especially in the early postnatal period, play a critical role in programming childhood AT and obesity [15]. Thus, accumulating evidence supports the view that early postnatal overnutrition alters epigenetic programming of AT, increasing the risk of obesity [16,17].

During the postnatal period, BF, in contrast to artificial FF, is found to have a protective effect against obesity [18,19,20]. Horta et al. [20] demonstrated that children (age 1–9 years), adolescents (age 10–19 years), and adults (≥20 years) who were breastfed as infants, have a reduction in the prevalence of overweight or obesity by 26% (95% CI: 21%; 32%), 37% (95% CI: 27%; 46%), and 12% (95% CI: 6%; 18%), respectively [20].

It is the intention of our review to provide molecular evidence that artificial FF, especially uncontrolled and increased protein intake by FF, compared to nature’s gold standard, BF, enhances adipogenesis and obesity risk caused by epigenetic deviations, suppressing wingless (Wnt) signaling and increasing adipogenic transcription. Exclusive BF obviously maintains the appropriate epigenetic axis during the postnatal period that reduces the risk of childhood overweight and obesity [21].

## 2. Current State of Knowledge

### 2.1. The Early Protein Hypothesis of Obesity Development

In 1992, Plagemann et al. [22] already observed in rats, through litter size reduction, that an increase in milk and milk protein access led to a higher risk of obesity, diabetes, and cardiovascular disease in adult rats due to early postnatal overfeeding effects. Obesity was correlated with basal hyperinsulinemia and increased systolic blood pressure in adult rats of small litters (postnatal milk excess) [22]. Kappeler et al. [23] demonstrated in litter size-modified mouse pups that both early postnatal undernutrition and overfeeding changes growth patterns through the developmental control of the somatotropic hormone axis. Recent evidence in rodent pups exposed to litter size reduction supports the concept that the obesity phenotype is promoted by early overnutrition [24,25,26]. In 1993, Dewey et al. [27] documented lower weight gain and percent body fat between 5 and 12 months of age in infants who received exclusive BF for 4–6 months compared to FF. The protein content of most infant and follow-up formulas during that time was greater than 2.25 and 2.6 g/100 kcal, respectively. Importantly, total energy intake at 3, 6, 9, and 12 months averaged 0.36, 0.34, 0.35, and 0.38 MJ/kg/day (85.9, 80.1, 83.6, and 89.8 kcal/kg/day) among BF versus 0.41, 0.40, 0.39, and 0.41 MJ/kg/day (98.7, 94.7, 93.6, and 98.0 kcal/kg/day) among FF infants, respectively. Protein intake was 66–70% higher in FF compared to BF during the first 6 months. Differences in energy and protein intakes were significant at 3, 6, and 9 months [28]. Based on protein intakes of breastfed infants, the estimated protein requirement was 1.98 g/kg/day during the first month of life and decreased rapidly to 1.18 g/kg/day by 4 to 5 months of age, then remained at about that level to age 1 year [29]. It is of critical concern that formula-fed infants can receive up to 2–3 times more protein than breastfed infants [9,30,31,32].

### 2.2. Epidemiological Evidence Supports the Early Protein Hypothesis

Two decades ago, the DARLING study [27] compared anthropometric parameters from 1 to 24 months of matched cohorts of infants either breastfed (n = 46) or formula-fed (n = 41) until 12 months or older. Formula-fed infants had larger skinfold thicknesses in later infancy (particularly 9–15 months) than did breastfed infants, and percent body fat (26.5% vs. 25.0%) was significantly higher from 5 to 24 months.

In a multicenter European study [33], 1138 healthy, formula-fed infants were randomly assigned to receive cow milk-based infant and follow-on formula with lower (1.77 and 2.2 g protein/100 kcal, respectively) or higher (2.9 and 4.4 g protein/100 kcal, respectively) protein contents for the first year. For comparison, 619 exclusively breastfed children were followed. The investigators demonstrated that a higher protein content in formula-fed infants was associated with higher weight (BMI) in the first 2 years of life [33].

Singhal et al. [30] reviewed a subset of children (n = 153 of 299 in study 1 and 90 of 246 in study 2) randomly assigned at birth to receive either a control formula or a nutrient-enriched formula (which contained 28–43% more protein and 6–12% more energy than the control formula) at 5–8 years of age. Fat mass, determined by impedance analysis or deuterium dilution, was lower in children receiving the control formula with lower protein content than in children receiving the protein-enriched formula. In nonrandomized analyses, faster weight gain in infancy was associated with greater fat mass in childhood [30].

A follow-up randomized trial by the European Childhood Obesity Trial Study Group [34] demonstrated that lower protein content in infant formula reduces BMI and obesity risk at school age. Children who received higher protein amounts exhibited a significantly higher BMI (by 0.51; 95% CI: 0.13, 0.90; *p* = 0.009) at 6 years of age. The risk of becoming obese in the higher-protein group was 2.43 times (95% CI: 1.12, 5.27; *p* = 0.024) that in the group raised on lower protein intake.

A systematic review considering randomized controlled trials until November 2014 concluded that current evidence is insufficient for assessing the effects of reducing the protein concentration in infant formulas on long-term outcomes, but, if confirmed, could be a promising intervention for reducing the risk of overweight and obesity in children [35].

Recently, the European Childhood Obesity Trial Study Group [36] confirmed that the risk for overweight and obesity at 8 and 11 years tends to be higher in the high-protein compared to low-protein and BF groups, with significant differences in the adjusted model for obesity at 8 years (adjusted OR 3.13; 1.23–7.99; *p* = 0.017) and for overweight at 11 years (adjusted OR 1.90; 1.12–3.21; *p* = 0.017). At 8 and 11 years, no child with late adiposity rebound (after 5 years) is considered as a child with obesity, but 94% of all children with obesity at 8 years (96% at 11 years) were very early rebounders (before 3.5 years). All children with obesity showed an adiposity rebound at younger than 4 years [36]. The majority of international experts in the field accept the association between excessive postnatal protein intake, accelerated infant growth, and increased risk of childhood obesity [31,37,38,39,40,41,42,43,44,45,46].

The question remains: Through which molecular mechanisms does excessive early protein intake enhance adipogenesis and increase the risk of obesity later in life?

The purpose of our review is to establish the molecular link by presenting recent findings and the latest insights into the epigenetics and molecular biology of postnatal adipocyte stem cell (ASC) commitment.

### 2.3. Increased mTORC1-S6K1 Signaling Induced by Formula Feeding

In 2015, Melnik [47] proposed that milk is the ideal nutrient system of mammalian evolution to control the appropriate activation of the mechanistic target of rapamycin complex 1 (mTORC1)-dependent translation by activating mTORC1’s downstream target, the kinase S6K1. mTORC1 is a nutrient- and growth factor-dependent kinase that coordinates cellular growth and anabolism while suppressing autophagy [48,49]. mTORC1 coordinates the translation of specific mRNAs crucial for cell growth and proliferation [50]. mTORC1 is activated by two key pathways: (1) Growth factor-stimulated PI3K-AKT signaling activating RAS homolog enriched in the brain (RHEB) [51,52,53]; and (2) amino acid (AA) availability for RAG GTPase-mediated activation of mTORC1 [54,55,56,57,58]. AA sufficiency and mTORC1 regulate the activation of S6K1 and phosphorylation of eukaryotic translation initiation factor 4E-binding protein 1 (EIF4EBP1) through a common effector mechanism [59]. The selective inhibition of S6K1 by AA withdrawal resembles the response to rapamycin, which prevents S6K1 reactivation by AAs, indicating that mTORC1 is required for the response to AAs [59].

In 2012, Melnik [60] suggested a mechanistic molecular pathway explaining early childhood obesity based on excessive leucine-mTORC1-S6K1 signaling resulting from high protein infant formula (AAs, leucine) intake, recently appreciated in the field of pediatric research [41].

#### 2.3.1. Insulin, IGF-1, and Branched-Chain Amino Acids Activate mTORC1-S6K1

The abundance of milk proteins by FF increases the plasma levels of essential branched-chain AAs (BCAAs) of the prototype leucine (LEU), and increases serum levels of insulin and insulin-like growth factor 1 (IGF-1), key upstream activators enhancing mTORC1-S6K1 signaling.

In a multicenter European study [61], 1138 healthy, cow milk-based formula-fed infants and follow-on formula-fed infants with lower protein (LP; 1.77 and 2.2 g protein/100 kcal) or higher protein (HP; 2.9 and 4.4 g protein/100 kcal) content were followed for the first year. Biochemical variables were measured at age 6 months in 339 infants receiving LP formula and 333 infants receiving HP formula compared to 237 breastfed infants. BCAAs, IGF-1, and urinary C-peptide/creatinine ratio, as measures of insulin release, were significantly (*p* < 0.001) higher in the HP group than in the LP and breastfed groups. The median IGF-1 total serum concentration was 48.4 ng/mL (25th, 75th percentile: 27.2; 81.8 ng/mL) in the HP group, 34.7 ng/mL (17.7; 57.5 ng/mL) in the LP group, and 14.1 ng/mL (5.1; 33.2 ng/mL) in the breastfed group, respectively. The urine C-peptide, a measure of insulin secretion, was 26.9 ng/mL (13.3; 45.6 ng/mL) in the HP group, 19.5 ng/mL (9.4; 34.6 ng/mL) in the LP group, and 9.3 ng/mL (3.5; 20.1 ng/mL) in the breastfed group, respectively.

Infant and follow-on formulas with lower protein content provided 119 and 154 mg leucine/100 mL, whereas infant and follow-on formula with higher protein content contained 197 and 308 mg leucine/100 mL, respectively. The leucine plasma levels of the HP group were 165 μmol/L (124; 212 μmol/L), 120 μmol/L (98, 143 μmol/L) of the LP group, and 106 μmol/L (90; 133 μmol/L) of the breastfed group, respectively [61].

It is important to note that leucine exerts stimulatory effects on insulin secretion of pancreatic β-cells [62,63] and regulates gene transcription and protein synthesis in pancreatic islet β-cells via both mTORC1-dependent and -independent pathways [62]. Furthermore, leucine enhances hepatic IGF-1 synthesis and serum levels of IGF-1 [64]. In fact, the consumption of cow milk and cow milk proteins increases insulin and IGF-1 serum levels, as shown in infants, prepubertal children, pubertal children, adolescents, and adults [61,65,66,67,68,69,70,71]. Most essential AAs, IGF-1, C-peptide, and urea increased significantly in both the HP and LP groups compared with the breastfed group [61].

According to Davis et al. [72], HBM on average contains 104 mg leucine/100 mL, the lowest among all mammals. The total leucine content of HBM depends on the lactation period: 153.7 mg/100 g (±63.2) days 5–11, 133.7 mg/100 g (±35.1) days 12–30, 130.3 mg/100 g (±33.5) months 1–2, 108.1 mg/100 g (±24.9) months 2–4, and 122.6 mg/100 g (±38.8) months 4–8, respectively [73], pointing to higher requirements of leucine supply during the early postnatal growth period. In comparison to BF, FF with higher protein content results in higher leucine intake. It has been recently confirmed by Slupsky et al. [74] that FF induces a rapid increase in circulating AAs, creatinine, and urea compared to breastfed infants. At 90 min and 120 min post-feeding, leucine and insulin serum levels were significantly elevated in infants receiving artificial formula compared to breastfed infants [74].

The nutrient-sensitive kinase mTORC1 and its downstream target S6K1 contribute to AA-induced insulin resistance [75,76]. In fact, it has been shown in the skeletal muscle of healthy men that combined hyperaminoacidemia and hyperinsulinemia increase S6K1 phosphorylation and inhibitory insulin receptor substrate-1 (IRS-1) phosphorylation at Ser312 and Ser636, whereas the mTORC1 inhibitor rapamycin partially inhibits this increase in mTORC1-mediated S6K1 phosphorylation and IRS-1 Ser312 and Ser636 phosphorylation [77].

It has also been demonstrated in newborn rhesus monkeys (Macaca mulatta) that FF exhibited higher serum levels of insulin and AAs [78]. Formula-fed rhesus infants weighed significantly more and were longer than their breastfed counterparts at all experimental time points starting from week 4 and continuing to 12 weeks of age (*p* < 0.05). Analysis of serum insulin concentrations showed a significant effect of diet, which was particularly pronounced in the early weeks of life (*p* < 0.01) [79]. A recent longitudinal study followed newborn rhesus monkeys either exclusively breastfed or fed regular formula or reduced protein formula, either supplemented or not with a mixture of free AAs. Despite the lower protein intake, these free AA-supplemented infants still exhibited a distinct FF-specific metabolic phenotype characterized by accelerated weight gain, higher levels of insulin and C-peptide, and elevated AAs, including BCAAs [80].

A recent study used the zebrafish (*Danio rerio*) as an experimental model to clarify whether early leucine stimulation can programmatically affect the mTORC1 signaling pathway, growth, and metabolism in later life, and to uncover potential mechanisms of epigenetic regulation [81]. Zebrafish larvae at 3 days post hatching (dph) were raised with 1.0% leucine from 3 to 13 dph during the critical developmental stage, then back to normal water for 70 days (83 dph). Growth performance in the early leucine programming group was increased, consistent with activation of mTORC1 signaling and high expression of genes involved in the metabolism of AAs and glycolipids. At 13 dph, the abundance of phosphorylated S6K1, S6, and growth factor receptor-bound protein 10 (GRB10) were increased in the larvae treated with leucine (*p* < 0.05). Furthermore, at 83 dph, compared with the control group, leucine programming also led to the higher abundance of phosphorylated S6K1 and S6 (*p* < 0.05). The mRNA expression of the *MTOR* gene in the leucine programming group was significantly higher compared to the control group at 13 dph and 83 dph, respectively (*p* < 0.05) [81].

Notably, decreased CpG methylation levels of *GRB10*, *EIF4E* and *MTOR* genes in the leucine programming group might contribute to their enhanced gene expression, demonstrating that leucine induces deviations in epigenetic regulation at the DNA level [81]. Recent evidence confirmed an epigenome-wide association of infant FF and changes in DNA methylation from birth to 10 years [82]. The sum of all changes in methylation from birth to age 10 years was significantly lower in the FF group compared to the BF group. Correspondingly, the number of CpGs with a methylation decline was 4.7% higher, reflecting 13,683 CpGs. Lower methylation related to exclusive FF and its adverse potential for the child’s development has been implicated [82].

Hoppe et al. [66] demonstrated that high intake of cow milk, but not meat, increases serum-insulin and insulin resistance in 8-year-old boys. Notably, early postnatal overfeeding of mice induced by litter size reduction led to reduced glucose tolerance later in life [22]. In accordance, FF at age 0–3 months was associated with greater insulin resistance during adolescence compared to exclusive BF [83]. Thus, excessive postnatal protein intake increasing the availability of BCAAs explains the early deleterious impact of FF by overactivation TORC1-S6K1 signaling [60], a postnatal disturbance of metabolic homeostasis during a vulnerable developmental window of epigenetic and metabolic programming, later maintained by high-protein complementary feeding [84] and a cow milk-based Western diet for school children, promoting insulin resistance [66], prediabetes [85], and T2DM in adult life [86,87,88,89], which are highly undesired early epigenetic and metabolic deviations initiated during early nutritional programming, paving the road to obesity and T2DM [90].

#### 2.3.2. *FTO* and Adipocyte Stem Cell Activation

The fat mass- and obesity-associated gene (*FTO*) is critically involved in the regulation of postnatal growth. Global germline loss of *FTO* in mice leads to postnatal growth retardation and a significant reduction in AT and lean body mass [91], underlining the fundamental developmental role of *FTO*. In contrast, single-nucleotide polymorphisms (SNPs) in intron 1 of *FTO* are associated with an increased risk of obesity [92,93]. Notably, the most prevalent SNP rs9939609 risk allele is associated with increased *FTO* mRNA levels [94,95]. In accordance, mice with two additional copies of *FTO* (FTO-4 mice) displayed increased adiposity [96]. Wåhlén et al. [97] reported that the AT level of *FTO* mRNA was increased in obesity (*p* = 0.002), was similar in subcutaneous and omental AT, and was higher in fat cells than in fat tissue (*p* = 0.0007). Remarkably, *FTO* was induced in preadipocytes at an early stage in the differentiation process (*p* = 0.004) [97]. Tews et al. [98] confirmed these observations and reported that subcutaneous fat-derived *FTO* in healthy lean and obese women significantly correlated with BMI. Obese women had a 25% higher *FTO* expression than women who were not obese. In preadipocytes isolated from subcutaneous AT and differentiating preadipocytes in culture, a decline in *FTO* expression could be observed [98]. The gradual downregulation of *FTO* during preadipocyte differentiation corresponds to studies in mice exhibiting a lower *FTO* expression in AT compared to a ~36% higher *FTO* expression (*p* = 0.005) in the stromal vascular fraction (SVF) [99], where mesenchymal stem cells (MSCs), adipocyte stem cells (ASCs), preadipocytes, resident immune cells including macrophages and T lymphocytes, fibroblasts, pericytes, and vascular endothelial cells reside [100,101,102,103,104,105].

Interestingly, Cheng et al. [106] reported that increased *FTO* expression via m^6^A RNA demethylation upregulates the expression of CD44, a cell-surface adhesion receptor and stem cell biomarker of ASCs. Using CRISPR/Cas9-mediated gene deletion and lentivirus-mediated gene re-expression, Weng et al. [107] discovered that deletion of CD44 promotes preadipocytes differentiation to adipocytes, whereas re-expression of CD44 abolishes this effect via suppressing *PPARG* expression [107]. These findings already point to a potential early impact of *FTO* on ASC regulation. The close vicinity of ASCs to the vascular systems may allow access to circulating compounds, including AAs, hormones (insulin, IGF-1), gut-derived metabolites, and circulatory exosomes and their miR cargo (Figure 1).

A recent study in obese and overweight male adolescents confirmed that high protein intake increases *FTO* expression after 18 weeks of intervention [108]. *FTO* expression is influenced by the availability of essential AAs [109]. *FTO* mRNA and protein levels are dramatically downregulated by total AA deprivation in mouse hypothalamic N46 cells, mouse embryonic fibroblasts (MEFs), and human HEK293 cells. The drop rate of *FTO* mRNA is faster than its rate of natural degradation, pointing to regulation at the transcriptional level, which is reversible upon AA replacement. Strikingly, this downregulation was seen only with essential AA deficiency but not with deficient nonessential AAs. These data suggest that *FTO* might function as a sensor of essential AA availability [109]. This implies that higher intake of essential AAs by FF enhances the overexpression of *FTO*. In fact, Cheshmeh et al. [110] demonstrated excessively overexpressed *FTO* in peripheral blood mononuclear cells (PBMCs) of formula-fed infants compared to exclusively breastfed infants at the age of 5 to 6 months. The group of exclusively breastfed infants exhibited the lowest level of *FTO* gene expression (3.39 ± 1.1) compared to the formula-fed group of infants (89.2 ± 19.3) (*p* < 0.001). A significant but intermediate increase in *FTO* gene expression compared to exclusive BF was also observed in the group of infants receiving mixed feeding (HBM and formula) (59.3 ± 9.3) [110], pointing to the importance of exclusive BF to maintain the low physiological *FTO* expression guaranteed by BF.

Intriguingly, Gulati et al. [111] described a role for *FTO* in the coupling of AA levels to mTORC1 signaling. Cells lacking *FTO* exhibit reduced activation of the mTORC1 pathway, slower mRNA translation, and increased autophagy [109,111]. Furthermore, they measured the status of mTORC1 signaling in *FTO*^−/−^ MEFs. Consistent with the reduced mRNA translation, phosphorylation of S6K1, the downstream target of mTORC1, was reduced in the *FTO*^−/−^ MEFs, indicating decreased basal levels of mTORC1 signaling in MEFs. Gulati and coworkers [111,112] predicted that *FTO* operates upstream of amino-acyl tRNA synthetases, especially leucyl-tRNA synthase (LARS), to connect leucine availability to mTORC1 activation [113]. Of note, LARS plays a critical role in AA-induced mTORC1 activation by sensing intracellular leucine concentration and initiating molecular events leading to mTORC1 activation [113]. LARS directly binds to RAG GTPase, the mediator of AA signaling to mTORC1, in an AA-dependent manner, and functions as a GTPase-activating protein (GAP) for RAG GTPase to activate mTORC1 [113,114]. During leucine signaling, LARS serves as an initiating “ON” switch via GTP hydrolysis of RAGD that drives the entire RAG GTPase cycle, whereas sestrin 2 functions as an “OFF” switch by controlling GTP hydrolysis of RAGB in the RAG GTPase–mTORC1 axis. The LARS–RAGD axis shows a positive correlation with mTORC1 activity [114]. LARS translocates to the lysosome on addition of leucine thereby activating mTORC1 [115].

Melnik [116] suggested that milk, via upregulation of *FTO*, promotes mTORC1 signaling. Recently, we provided a mechanistic link connecting *FTO* activity and upregulation of LARS promoting RAGD-mTORC1 signaling [117]. *FTO* exerts demethylase activity directed at single-stranded N6-methyladenosine (m^6^A) of RNA and N6-methyldeoxyadenosine (^6^mA) of DNA. *FTO* removes methyl groups from RNA m^6^A marks [118,119], directly promoting obesity through m^6^A RNA modifications [117,120,121,122,123,124,125,126]. Furthermore, FTO-mediated RNA demethylation occurs to m^6^A_m_ in mRNA and snRNA as well as m^1^A in tRNA [127].

Generally, m^6^A demethylation increases the mRNA expression of adipogenic transcription factors [117,122]. Specifically, *FTO* upregulates the expression of sterol regulatory element-binding transcription factor 1 (*SREBF1*) [128]; peroxisome proliferator-activated receptor-γ (*PPARG*) [129]; Runt-related transcription factor 1, translocated to 1 short form (*RUNX1T1-S*) [120]; CCAAT/enhancer-binding protein α (*CEBPA*) [129]; CCAAT/enhancer-binding protein β (*CEBPB*) [130]; CCAAT/enhancer-binding protein δ (*CEBPD*) [130]; and MYC proto-oncogene (*MYC*) [131,132].

Of note, MYC is an early response regulator of human adipogenesis in ASCs [132] and promotes multipotent MSCs to the adipogenic lineage [132]. Recent evidence indicates that MYC activates the expression of activating transcription factor 4 (*ATF4*) [133].

It is important to note that FTO-mediated m^6^A RNA demethylation increases the expression of *ATF4* [134,135]. *ATF4* is a key transcriptional activator of LARS expression [136]. The CEBP-ATF response element (CARE) location reveals two distinct *ATF4*-dependent, elongation-mediated mechanisms for transcriptional induction of aminoacyl-tRNA synthetase genes [137].

Moreover, *FTO* decreases gene expression of TSC complex subunit 1 (*TSC1*), while knockdown of *FTO* increases the mRNA level of *TSC1* [138]. The *TSC1*/*TSC2* complex is a critical inhibitory checkpoint of growth factor (insulin/IGF-1)-induced mTORC1 activation [139]. FTO-mediated suppression of *TSC1* may enhance RHEB-mediated activation of mTORC1. Thus, AA-mediated upregulation of *FTO* enhances both AA- and growth factor-mediated signaling towards mTORC1. Intriguingly, Torrence et al. [140] recently demonstrated that *ATF4* is synthesized by activated mTORC1. mTORC1 also increases mRNA and protein expression of PPARγ and SREBFs [141], two master transcriptional regulators of adipocyte differentiation and lipogenesis [142,143]. Remarkably, the transcription of C/EBPβ and PPARγ are activated by *ATF4* [144]. Whereas overexpression of *ATF4* in 3T3-L1 cells enhanced adipogenesis, small-interfering *ATF4* blocked conversion of preadipocytes to adipocytes [144] emphasizing the key role of *ATF4* in early adipocyte differentiation.

Of note, *ATF4* co-localizes with CCCTC-binding factor (CTCF) in the promoters of key adipogenic genes, including *CEBPD* and *PPARG*, and co-regulates their transactivation [145]. Thus, *ATF4* and CTCF work cooperatively to control adipogenesis and adipose development via orchestrating transcription of the adipogenic genes *CEBPB*, *CEBPD*, and *PPARG* [145,146]. Furthermore. *ATF4* has been shown to upregulate the expression of *SREBF1* and *CEBPB* [133]. It has also been demonstrated in the zebrafish that *ATF4* overexpression accelerated adipocyte differentiation via C/EBPβ and PPARγ expression [146]. In accordance, overexpression of *FTO* in the zebrafish resulted in fat accumulation, and upregulation of PPARγ and C/EBPα, as well as a decrease in the global m^6^A level in larvae [147]. Cohen et al. [148] recently uncovered a novel mechanism regulating transcription in human MSCs adipogenically primed by confluence. Prior to adipogenesis, confluency promotes heterodimer recruitment of the bZip transcription factors C/EBPβ and *ATF4* to a non-canonical CEBP DNA sequence, whereas *ATF4* depletion decreases both cell-density-dependent transcription and adipocyte differentiation [148].

CTCF has also been shown to bind an enhancer region of *FTO* promoting *FTO* expression [149]. Deletion of the CTCF site in *FTO* in mice resulted in normal food intake and an inability to become obese when ancestrally exposed to bisphenol A [149].

*ATF4*-deficient mice exhibit increased energy expenditure, enhanced lipolysis, upregulation of uncoupling protein 2 (UCP-2) and β-oxidation genes, and decreased expression of lipogenic genes in white adipose tissue (WAT) [150]. In addition, adult-onset agouti-related peptide neuron-specific *ATF4* knockout (AgRP-*ATF4* KO) mice are lean, and exhibit improved insulin and leptin sensitivity and reduced hepatic lipid accumulation [151].

*Drosophila melanogaster* mutant flies with insertions at the *ATF4* locus exhibit reduced fat content, increased sensitivity to starvation, and lower circulating carbohydrate levels [152]. *ATF4* null mice are also lean and resistant to age-related and diet-induced obesity. Several aspects of the *ATF4* mutant phenotype resemble mice with mutations in components of the mTOR pathway. Furthermore, *ATF4* null mice have reduced expression of genes that regulate intracellular AA concentrations and lower intracellular concentration of AAs, key regulators for mTORC1 activation. In accordance, *ATF4* mutants have reduced S6K activity in liver and AT [152].

In summary, increased AA intake via upregulation of *FTO* stimulates *ATF4*-LARS-mTORC1-S6K1 signaling and *ATF4*-C/EBPβ-PPARγ transcriptional upregulation, critical events promoting MSC commitment to the adipocyte lineage and subsequent ASC differentiation.

#### 2.3.3. Breastfeeding Counteracts the *FTO* rs9939609 Obesity Risk Allele

Exclusive BF acts antagonistically to the *FTO* gain-of-function rs9939609 risk allele [153,154]. In particular, Wu et al. [153] studied 5590 children from the British Avon Longitudinal Study of Parents and Children (ALSPAC) cohort and modelled their longitudinal BMI profiles with mixed-effects models from birth to 16 years of age. They also looked at their ages at adiposity peak (AP), adiposity rebound (AR), and BMI velocities in relation to the *FTO* gene variant and exclusive BF. Importantly, a longer duration of exclusive BF (at least 5 months) has substantial impact on BMI growth trajectories among children carrying the FTO-adverse variant by modulating the age at AP, age at AR, and BMI velocities. Exclusive BF acts antagonistically to the *FTO* rs9939609 risk allele by the age of 15. The predicted reduction in BMI after 5 months of exclusive BF was 0.56 kg/m^2^ (95% CI 0.11; 1.01; *p* = 0.003) and 1.14 kg/m^2^ (95% CI 0.67; 1.62; *p* < 0.0001) in boys and girls, respectively.

Horta et al. [154] assessed the association of BF with body composition at 30 years, among subjects who were prospectively followed since birth in a southern Brazilian city. They evaluated whether BF moderated the association between the rs9939609 variant in the *FTO* gene and adiposity. They showed that at 30 years, total and predominant BF were positively associated with lean mass index and inversely with visceral fat thickness. Among subjects with BF for <1 month, all outcomes showed monotonically increasing values with additional copies of the A allele in the *FTO* genotype (rs9939609). Associations among subjects with BF for one month or longer tended to be in the same direction but showed lower magnitude and were less consistent for all outcomes (interactions had *p* ≤ 0.05 for BMI, fat mass index, and waist circumference). Thus, even among young adults, BF moderates the association between the *FTO* variant rs9939609 and body composition [154].

As demonstrated by Cheshmeh et al. [110], 5–6-month-old infants exclusively BF exhibit significantly decreased levels of *FTO* expression in PBMCs compared to mixed feeding or exclusive FF. These results may be explained by less total protein and less BCAA intake of breastfed infants compared to formula-fed infants.

#### 2.3.4. NADP Increases *FTO* Activity in Preadipocytes

Wang et al. [155] demonstrated that *FTO* enzymatic activity is upregulated by nicotinamide adenine dinucleotide phosphate (NADP), while deletion of *FTO* blocked NADP-enhanced adipogenesis in 3T3-L1 preadipocytes. NADP directly binds to the *FTO* protein and increases its enzymatic activity, promoting RNA m^6^A demethylation and early adipogenesis [155]. Compared to BF, FF leads to excessive intake of tryptophan (TRP) [31] and its downstream metabolite kynurenine (KYN) [156], which, after conversion to quinolinic acid and nicotinamide adenine dinucleotide (NAD), is further metabolized to nicotinamide adenine dinucleotide phosphate (NADP) [157]. Therefore, early high protein and TRP intake may overstimulate NADP-stimulated enzymatic activity of *FTO* of adipocyte progenitor cells (APCs). Furthermore, the *FTO* rs9939609 A allele is likely to influence the conversion of TRP to kynurenine [158].

#### 2.3.5. *FTO* Regulation by MicroRNAs

HBM and formula not only differ in the amounts of protein and AA (BCAA) intake but also in the content of exosome- and milk fat granule-derived microRNAs (miRs), which are deficient in formula [159,160]. *FTO* expression can be suppressed by FTO-targeting miRs. For instance, miR-30b-5p has been shown to downregulate *FTO* expression in zebrafish, resulting in a reduction of lipogenesis [147]. MiR-30b/c-5p also induces thermogenesis and promotes the development of beige fat by targeting receptor-interacting protein 140 (RIP140) [161]. MiR-30b-5p is a major exosomal miR of human colostrum [162] and mature HBM [163,164]. Notably, *FTO* deficiency promotes thermogenesis and the transition of white-to-beige adipocytes via YTHDC2-mediated translation and increased expression of hypoxia-inducible factor-1α (HIF-1A) [165]. Furthermore, *FTO* deficiency in mice modifies gene and miR expression involved in brown adipogenesis and browning of WAT [166]. *FTO* deficiency upregulates uncoupling protein 1 (UCP-1) and subsequently enhances mitochondrial uncoupling and energy expenditure, resulting in the induction of the brown adipocyte phenotype [167].

Notably, miR-22-3p is another exosomal milk miR targeting *FTO* [168]. Remarkably, miR-22-3p is significantly overexpressed in HBM exosomes (HBMEs) of mothers delivering preterm infants [169,170]. This miR not only promotes intestinal cell proliferation [169,170], but also upregulates the development of brown adipose tissue (BAT) in response to cold exposure and during brown preadipocyte differentiation [171]. In accordance, bone marrow MSCs (BMSCs) secrete miR-22-3p-enriched exosomes that negatively target *FTO* thereby promoting osteogenic differentiation [172]. Further evidence underlines that growth/differentiation factor 11 (GDF11)-upregulated C/EBPa enhances *FTO* expression [173]. *FTO* via m^6^A demethylation stimulates the expression of *PPARG*, shifting the MSC fate towards the adipocyte lineage but inhibiting bone formation [173,174]. In accordance, it has been shown that the expression of *FTO* is upregulated during adipocyte differentiation of BMSCs [173,174], whereas *FTO* expression is downregulated during osteoblast differentiation [174,175].

Notably, Li et al. [176] showed that miR-149-3p via *FTO* repression also inhibits adipogenic lineage but potentiates osteogenic lineage differentiation. Kupsco et al. [177] detected miR-149-3p as a member of the top 10 miRs in a miR cluster of HBM extracellular vesicles (EVs). MiR-22-3p operates in BAT by targeting HIF-1α inhibitor (*HIF1AN*), which is a key inhibitor of glycolysis, thereby promoting thermogenesis by enhancing the activity of HIF-1α, the master transcription factor of glycolysis and thermogenesis [178]. *HIF1AN* mRNA is also a target of miR-148a-3p [178], which is the dominant miR of HBM and HBMEs [179,180,181,182,183] and is also upregulated in HBMEs of mothers delivering preterm babies [169]. 3T3-L1 preadipocytes treated with siRNA targeted against *FTO* prior to in vitro adipogenesis impaired the ability of 3T3-L1 cells to develop into mature adipocytes, pointing to a key role of *FTO* in preadipocyte differentiation [184]. It is of critical importance to note that *FTO* knock-down significantly upregulated Wnt genes including *WNT10B*, which was previously reported to inhibit adipogenesis [185,186]. Importantly, knockdown of *FTO* prior to differentiation has been confirmed to impair adipogenesis in 3T3-L1 adipocytes and human ASCs [130]. Thus, HBME-derived miRs may calibrate the appropriate magnitude of postnatal *FTO* expression, thereby controlling ASC differentiation and adjusting the balance between WAT and BAT development, a meaningful adaptation to the metabolic requirements (thermogenesis) and maturation state of the newborn infant (very preterm, preterm, term). The miR deficiency of formula would thus result in higher levels of *FTO* expression, enhancing ASC differentiation and increasing WAT but impairing BAT development.

The expression of *FTO* is activated by direct binding of zinc finger protein 217 (ZFP217) to the *FTO* promoter [187,188]. ZFP217 (encoded by *ZNF217*) positively regulates the m^6^A epitranscriptome involved in adipogenesis and interacts with YTHDF2 to maintain m^6^A demethylation activity of *FTO* [187]. Loss of *ZNF217* retards adipose differentiation and enhances m^6^A modification during adipogenesis. *ZNF217* knockdown significantly blocked adipogenesis, as indicated by a decreased level of Oil Red O staining as well as lower expression of key adipogenic factors like PPARγ, adipocyte protein 2 (aP2), lipoprotein lipase (LPL), and adiponectin [187]. Knockdown of *ZNF217* inhibits mitotic clonal expansion (MCE) and adipogenesis [188,189]. Mice deficient in ZFP217 resist HFD-induced obesity by increasing energy metabolism [190].

ZFP217 not only directly binds to DNA but also acts as a bridge that recruits a transcriptional repressor complex, regulating the transcriptional functions of target genes [191,192]. Intriguingly, ZFP217 interacts with and increases the activity of the lysine methylase enhancer of zeste homolog 2 (*EZH2*) [193]. *EZH2* is the principal H3K27 methylase generating histone H3 lysine 27 methylation (H3K27me3) resulting in polycomb-group silencing and suppression of *WNT* genes [194,195,196,197,198,199]. ZFP217 regulates the C-terminal binding protein 2 (CTBP2)-mediated recruitment of the nucleosome remodeling and deacetylation (NURD) complex and polycomb repressive complex 2 (PRC2) to active embryonic stem cell (ESC) genes, subsequently switching the H3K27ac to H3K27me3 during ESC differentiation for active gene silencing [200]. In accordance, ZFP217 depletion retards ESC differentiation [200]. The canonical Wnt pathway is critical for ESC pluripotency and aberrant control of β-catenin leads to failure of exit from pluripotency and lineage commitments [201,202]. Inhibition of *EZH2* activity and knockdown of *EZH2* gene expression in human MSC resulted in decreased adipogenesis [203]. In fact, Wang et al. [204] convincingly demonstrated that *EZH2* and its H3K27 methyltransferase activity are required for adipogenesis. *EZH2* directly represses *WNT1*, *WNT6*, *WNT10A*, and *WNT10B* genes in preadipocytes and during adipogenesis [204]. In contrast, deletion of *EZH2* eliminates H3K27me3 on *WNT* promoters and derepresses *WNT* expression, which leads to activation of Wnt/β-catenin signaling and inhibition of adipogenesis [204].

HBME miR-148a-3p/miR-200bc-3p/miR-17-5p-mediated suppression of *ZNF217* [205] may attenuate ZFP217-stimulated *FTO* expression and ZFP217-activated *EZH2* activity. This may result in suppressed *FTO* activity but enhanced Wnt signaling, which is a potentially important postnatal gene regulatory epigenetic network that is absent in artificial formula.

#### 2.3.6. Systemic Milk MicroRNA Uptake Determined by Postnatal Intestinal Permeability

It is important to consider that there might be a critical time window for preferred systemic HBME miR uptake and miR-mediated regulation of ASC development. Intestinal permeability is highest directly during the first week after birth [206]. In 3–6-day-old human neonates, intestinal permeability decreases in both term and preterm neonates [206,207]. In the first postnatal month, intestinal permeability of preterm infants significantly decreases for infants receiving BF versus FF in a dose-related manner [208]. Weil et al. [209] performed cross-species profiling of miRs via deep sequencing and utilized dietary xenobiotic taxon-specific milk miRs (xenomiRs) as tracers in human and porcine neonates, followed by functional studies in primary human fetal intestinal epithelial cells using adenovirus-type 5-mediated miR gene transfer. Milk-derived miRs survived the gastrointestinal passage in human and porcine neonates. Bovine-specific miRs accumulated in intestinal cells of preterm piglets after enteral feeding with bovine colostrum/formula. In piglets, colostrum supplementation with cel-miR-39-5p/-3p resulted in increased blood concentrations of cel-miR-39-3p and argonaute RISC catalytic component 2 (AGO2) loading in intestinal cells, suggesting the possibility of vertical transmission of miRs from milk through the neonatal digestive tract [209]. Recently, Swanson et al. [210] exposed young piglets after the weaning period with excessive amounts of bovine milk. Remarkably, cow milk consumption increased the number of ASCs in subcutaneous WAT but did not affect adipogenic differentiation of ASCs. Bovine exosomal miRs could not be detected in porcine plasma, suggesting that miRs are not vertically transferred from bovine milk exosomes after the piglet’s weaning period [210]. Remarkably, cow milk intake increased the number of ASCs in this experimental setting, pointing to miR-independent milk factors enhancing ASC numbers after weaning.

#### 2.3.7. The Role of S6K1 in Early Adipogenesis

Bar Yamin et al. [211] studied the long-term effects of a regular chow diet with or without supplementation of commercial cow milk in newly weaned mice. They detected a significant increase of phosphorylated S6K1 (pS6K1) in the WAT of the milk group, most likely explained by milk protein-induced overactivation of FTO-mTORC1-S6K1 signaling [47,60,115]. Metformin, the most commonly used anti-diabetic drug, is an inhibitor of mTORC1 [212,213]. Chinnapaka et al. [214] recently demonstrated that metformin improves the stemness of ASCs, reducing their rate of proliferation and adipocyte differentiation by decreasing mTORC1 signaling and reducing levels of pS6K1 [214]. In fact, metformin and the mTORC1 inhibitor rapamycin downregulate the expression of pS6K1 and increase the stemness of ASCs [214]. Rapamycin treatment reduces clonal expansion, C/EBPα expression, and 3T3-L1 preadipocyte differentiation [215]. In addition, rapamycin not only prevents adipocyte differentiation by decreasing adipogenesis and PPARγ expression but also downregulates insulin action in adipocytes, implying that mTORC1 plays important roles in adipogenesis and insulin action [216]. In accordance, AT-specific knockout of raptor (raptor(ad^−/−^), the essential functional component of mTORC1, substantially reduced AT in raptor(ad^−/−^) mice and protected these mice against diet-induced obesity [217]. The WAT of raptor(ad^−/−^) mice displayed enhanced expression of mitochondrial uncoupling genes characteristic of BAT, and leanness was attributed to elevated energy expenditure [217]. These data fit well to previous observations of Carnevalli et al. [218], who provided evidence in mice that the mTORC1 downstream target S6K1 plays a critical role in the commitment of ESCs to early adipocyte progenitors. Earlier studies demonstrated that *S6K1*^−/−^ mice have reduced AT mass and increased energy expenditure, and are resistant to diet-induced obesity [219]. Eukaryotic translation initiation factor 4E-binding proteins (4EBPs), which repress translation by binding to eIF4E, are downstream effectors of mTORC1. *4EBP1*^−/−^/*4EBP2*^−/−^ mice displayed increased sensitivity to diet-induced obesity due to an acceleration in adipogenesis associated with hyperactivated S6K1 [220].

Carnevalli et al. [218] concluded that a lack of S6K1 impairs the generation of de novo adipocytes when mice are challenged with an HFD, consistent with a reduction in early adipocyte progenitors. Hyperactivation of *FTO* by FF [110] resulting from excessive protein intake [117] with resultant FTO-mediated overstimulation of mTORC1-S6K1 signaling may thus promote the proliferation of early adipocyte progenitors. Accumulated evidence underlines a critical impact of *FTO* on early steps of adipogenesis [221]. In fact, knockdown of *FTO* decreased adipogenesis in 3T3-L1 preadipocytes and human ASCs [120,130,184] as well as in porcine preadipocytes [222]. MEFs from *FTO* knockout (FTO-KO) mice exhibited reduced adipogenic potential, whereas overexpression of *FTO* led to an enhanced adipogenic program in primary murine preadipocytes, 3T3L1 preadipocytes, and porcine preadipocytes [120,222,223]. In particular, primary adipocytes and MEFs derived from *FTO* overexpressing (FTO-4) mice exhibited increased potential for adipogenic differentiation, while MEFs derived from FTO-KO mice showed reduced adipogenesis [223].

#### 2.3.8. S6K1, *FTO*, C/EBPβ Enhance Mitotic Clonal Expansion

As demonstrated in S6K1^−/−^ mice, S6K1 is critically involved in the commitment of ESCs to early adipocyte progenitors [218]. Importantly, Merkestein et al. [223] provided the first evidence for the involvement of *FTO* in early adipogenesis by regulating MCE. Knockdown of *ZNF217*, an activator of *FTO* expression [187], also inhibits MCE [188]. Conversely, overexpression of *FTO* significantly reduced N6-methyladenosine (m^6^A) levels and promoted proliferation and differentiation of chicken preadipocytes [224]. MCE is a synchronous process and a prerequisite for the differentiation of 3T3-L1 preadipocytes into adipocytes [225,226].

Tang et al. [227] reported that C/EBPβ/δs are expressed early in the differentiation program, but are not immediately active. After a lag phase, C/EBPβ/δs become competent to bind to the C/EBP regulatory element in the *CEBPA* promoter, resulting in C/EBPα-induced transcriptional activation of numerous adipocyte genes. As C/EBPβ/δs acquire binding activity, they become localized to centromeres as preadipocytes synchronously enter the S phase at the onset of MCE. Localization to centromeres occurs through C/EBP consensus-binding sites in centromeric satellite DNA. C/EBPα, which is antimitotic, becomes centromere-associated much later in the differentiation program as MCE ceases and the cells become terminally differentiated [227]. Further evidence in MEFs supports that CEBPβ is a prerequisite for MCE in the adipocyte differentiation program [228]. Gap junctional communication is able to inhibit MCE via modulating C/EBPβ expression [229].

Recently, Wang et al. [124] demonstrated that forced expression of *FTO* promoted C/EBPβ protein levels. In fact, *FTO* promoted autophagy and facilitated adipogenesis through mediating *CEBPB* expression. In contrast, adipocyte-selective *FTO* knockdown inhibited ATG5 and ATG7-dependent autophagy and *CEBPB* expression in mice. Thus, FTO-dependent regulation of ATG5 and ATG7-CEBP/β signaling modulates adipose tissue expansion [124].

Martin Carli et al. [130] demonstrated that *FTO* is permissive for adipogenesis and induces C/EBPβ-driven transcription and expression of *CEBPD* in human and murine ASCs. In contrast, *FTO* knockdown decreased the number of 3T3-L1 cells that differentiated into adipocytes as well as the amount of lipid per mature adipocyte. This effect on adipocyte programming was conveyed, in part, by modulation of C/EBPβ-regulated transcription. In addition, *FTO* affects *CEBPD* transcription by demethylating DNA N6-methyldeoxyadenosine (^6^mA) in the *CEBPD* promoter. In FTO-knockdown 3T3-L1 preadipocytes, *CEBPD* expression is decreased. In accordance, *FTO* knockdown in human ASCs decreased *CEBPB* and *CEBPD* expression, pointing to an early permissive role of *FTO* in adipogenesis [130]. As the cells cross the G_1_/S checkpoint, C/EBPβ acquires DNA-binding activity and initiates a cascade of transcriptional activation that culminates in the expression of adipocyte proteins [225]. Hirayama et al. [230] showed that *FTO* regulates G_1_ phase progression by modulating m^6^A modification of *cyclin D1*, supporting the critical role of *FTO* for MCE. Dominant-negative CEBP disrupts MCE and differentiation of 3T3-L1 preadipocytes [231].

#### 2.3.9. *FTO* Upregulates Flotillin 2, Promoting PI3K/AKT/mTORC1/S6K1 and ECM Signaling

Gulati and coworkers [109,111,112] have already provided evidence that *FTO* operates upstream of mTORC1, promoting LARS-mediated AA signaling towards mTORC1 activation. In addition, Jiao et al. [232] showed that *FTO* stimulates the proliferation and differentiation of preadipocytes involving upstream PI3K/AKT activation. *FTO* knockdown inhibits AKT phosphorylation, and PI3K inhibition by Wortmannin inhibits p-AKT in *FTO* overexpressed 3T3-L1 cells [232]. It is worth noting that *FTO* regulates the proliferation of 3T3-L1 cells in the precursor status, and even earlier during adipogenesis (24 h after adipogenic induction) [232]. In accordance, Tews et al. [98] observed an early upregulation of *FTO* in preadipocytes but downregulation of *FTO* during adipocyte differentiation. In gastric and breast and cancer cells, increased *FTO* expression has been related to PI3K/AKT activation, whereas *FTO* suppression reduced PI3K/AKT signaling [233,234]. The mechanisms through which *FTO* affects AKT phosphorylation are not known. However, in cervical cancer cells, p-AKT has been associated with increased expression of flotillin 2 encoded by the gene *FLOT2* [235]. Overexpression of *FLOT2* exacerbates the proliferation and epithelial/mesenchymal transition of cervical cancer cells [235], while silencing of *FLOT2* attenuated p-AKT in breast cancer cells [236]. Zhang et al. [237] reported that flotillin 2 contributes to cancer aggressiveness in diffuse large B-cell lymphoma by activating the PI3K/AKT/mTORC1 signal pathway. Intriguingly, *FTO* demethylates m^6^A modifications in *FLOT2* mRNA, upregulating *FLOT2* expression subsequently increasing the expression levels of p-PI3K, p-AKT, and p-mTOR [237]. Flotillin 2, like flotillin 1 ( *FLOT1*), is a highly conserved protein isolated from caveolae/lipid raft domains tethering growth factor receptors linked to signal transduction pathways [238]. shRNA knockdown of *FLOT1/2* and adaptor-related protein complex 2 subunit α1/2 (AP2A1/2) reduced IGF1R association with clathrin, internalization, and pathway activation by more than 50% of p-IGF1R and p-AKT [239]. In contrast to *FLOT2*, an increase in m^6^A modifications in *FLOT1* mRNA enhanced *FLOT1* mRNA expression [240]. Flotillin-1, syntaxis 13, and ATP-binding cassette transporter 1 (ABCA1) were identified as phagosomal proteins, indicating their involvement in ABCA1-mediated lipid efflux [241].

The SVF-derived extracellular matrix (ECM) modulates differentiation capacity of ASCs. Intriguingly, Grandl et al. [242] identified flotillin 2 as a protein enriched in pro-adipogenic ECM, orchestrating ECM to preadipocyte signaling. Remarkably, in pro-adipogenic ECMs, *FLOT2* exhibited a roughly 2-fold increased expression [242]. Conversely, siRNA mediated knockdown of *FLOT2* in 3T3-L1 preadipocytes abolished the stimulatory effect of vitamin C on adipocyte differentiation [242].

Of note, *FLOT2* mRNA is a predicted target of miR-148a-3p [243], suggesting a potential involvement of HBMEs in attenuating *FTO*/*FLOT2* expression. Recent evidence indicates that stem cell exosomes deliver *FTO* [244,245,246]. Specifically, BM-MSCs [245] and neuronal stem cells [246] transport *FTO* via exosomes to adjacent recipient cells. Therefore, it is conceivable that FTO-enriched exosomes contribute to *FTO* spreading to adjacent ASCs in exosome-regulated adipogenesis. A decade ago, Sebert et al. [247] already discussed the *FTO* gene in the early-life determination of body weight, body composition, and energy balance.

Thus, FTO-mediated upregulation of *FLOT2* may increase IGF1R-PI3K-AKT-mTORC1-S6K1 signaling and pro-adipogenic ECM modifications, promoting early steps of adipogenesis.

### 2.4. FTO-Mediated Wnt Suppression and BMP Activation Promote ASC Commitment

#### 2.4.1. FTO-mTORC1-S6K1-*EZH2*-Wnt-Signaling

Adipocytes are thought to originate from multipotent MSCs located in the AT VSF [100,101,102,103,104,105,248]. Accumulated evidence supports the inhibitory role of canonical Wnt signaling in the initial stages of adipogenesis [185,186,204,249,250]. Convincingly, it has been shown that the addition of Wnt10b anti-sera to 3T3-L1 preadipocyte media promotes adipocyte differentiation [251]. Furthermore, *WNT10B* expression, which is highest in preadipocytes, declines rapidly after induction of differentiation [185,252]. There is now compelling evidence that the canonical Wnt pathway regulates MSC fate determination in vivo in humans and mice. Reduced Wnt signaling is required to induce MSCs to undergo adipogenesis [249,253]. The Wnt signaling network might serve to tightly regulate AT expansion, and thus the susceptibility to obesity [249]. Wnt10b suppresses adipogenesis of WAT [185] and *WNT10B* overexpression also inhibits brown adipogenesis [254]. Early activation of canonical Wnt signaling hinders brown adipogenesis [254]. Therefore, Wnt signaling may have a significant impact on regulating of body fat distribution and, to some extent, susceptibility to obesity [249]. Moreover, accumulating evidence supports the notion that Wnt signaling is pivotal in determining the fate of multipotent MSCs to differentiate into preadipocytes [250].

Wnt signaling is epigenetically regulated and controlled by histone modifications [255]. Notably, an enrichment of histone H3K27 methyltransferase *EZH2* has been detected on *WNT* genes. *EZH2* directly represses *WNT1*, *WNT6*, *WNT10A*, and *WNT0B* genes in preadipocytes and during adipogenesis [204]. In contrast, deletion of *EZH2* eliminates H3K27me3 on *WNT* promoters and derepresses *WNT* expression, which leads to activation of Wnt/β-catenin signaling and inhibition of adipogenesis. Thus, H3K27 methyltransferase *EZH2* directly represses *WNT* genes to facilitate adipogenesis, suggesting that trimethylation on H3K27 plays opposing roles in regulating *WNT* expression [204].

Hemming et al. [203] demonstrated that *EZH2* and lysine demethylase 6a (KDM6a) change transcript levels during differentiation of multipotent human bone marrow-derived MSCs. Whereas enforced expression of *EZH2* in MSC promotes adipogenic differentiation and inhibits osteogenic differentiation, lysine demethylase 6A (KDM6A), which removes repressive trimethylation of histone H3 at lys27 (H3K27me3), inhibits adipogenesis and promotes osteogenic differentiation. Inhibition of *EZH2* activity and knockdown of *EZH2* gene expression in human MSCs result in decreased adipogenesis and increased osteogenesis. An important epigenetic switch, centered on H3K27me3, dictates MSC lineage determination [203]. In fact, Wu et al. [256] recently demonstrated that *EZH2*-deficient mice have a leaner phenotype and less WAT, enabling them to tolerate cold stimulation and resist obesity and insulin resistance induced by HFD. Remarkably, the *EZH2* inhibitor GSK126 can inhibit the differentiation of MEFs into white adipocytes but promotes their differentiation into brown/beige adipocytes [256].

The mTORC1-dependent phosphorylation of S6K1 at T389 is essential for its nuclear localization and exclusively hyperphosphorylated S6K1 can be found in the nucleus [257]. Intriguingly, Yi et al. [258] found a molecular link between overstimulated S6K1 and suppression of Wnt signaling promoting early steps of adipogenesis. Adipogenic stimuli trigger nuclear translocation of pS6K1, leading to histone 2B (H2BS36) phosphorylation and recruitment of *EZH2* to histone 3 (H3), mediating H3K27 trimethylation. This blocks *WNT* gene expression and consecutively upregulates C/EBPα and PPARγ, thus promoting adipogenesis. Consistent with this finding, WAT from S6K1-deficient mice exhibits no detectable H2BS36 phosphorylation or H3K27 trimethylation, while both responses are highly elevated in obese humans or in mice fed an HFD. These findings define an S6K1-dependent epigenetic mechanism in early adipogenesis, contributing to the promotion of obesity [258]. Thus, S6K1 plays a crucial role in controlling the downstream epigenetic and transcriptional programs required for the commitment of MSCs to the adipocytic lineage [258,259] (Figure 2).

Wan et al. [260] recently reported that macroH2A1.1 (mH2A1.1), a variant of histone H2A, was upregulated during adipocyte differentiation in 3T3-L1 cells and in the WAT of obese mice. Ablation of mH2A1.1 activated the Wnt/β-catenin signaling pathway, while overexpression of mH2A1.1 showed opposite effects. Of note, mH2A1.1, in cooperation with *EZH2*, regulated Wnt/β-catenin signaling, resulting in the accumulation of H3K27me2 and H3K27me3 on the promoters of *WNT* genes [260].

The absence of S6K1 protects mice against age- and HFD-induced obesity [212] and reduces MCE of preadipocytes [221]. Conversely, AA sufficiency [54,55,56,57,58,59,60], activated *FTO* [111,112], and mTORC1 activation [261,262,263] converge in the activation of S6K1 (pS6K1). Inhibition of mTORC1 with either metformin or rapamycin suppressed downstream pS6K1 levels in human ASCs [214]. Metformin has also been shown to suppress *FTO* protein expression in mice and exert an inhibitory effect on adipogenesis during the early stage of adipocyte differentiation targeting MCE [129]. Notably, the *FTO* promoter is also under control of Wnt signaling [264]. Wnt signaling induces the binding of *EZH2* to β-catenin. This *EZH2*/β-catenin complex binds to the LEF/TCF-binding elements at the promoter region of *FTO*, where *EZH2* enhances H3K27me3 and thereby inhibits *FTO* expression [264]. Thus, mTORC1-S6K1-mediated suppression *WNT* gene expression may further augment the expression of *FTO* and the adipogenic FTO-mTORC1-S6K1 signaling (Figure 2).

Interestingly, fisetin (3,7,3′,4′-tetrahydroxyflavone), present in vegetables and fruits, has been shown to prevent HFD-induced obesity through the regulation of mTORC1 [265,266,267]. Fisetin treatment of preadipocytes reduced the phosphorylation of S6K1 and mTORC1 in a time- and concentration-dependent manner [266]. In 3T3-L1 cells, fisetin inhibited the phosphorylation of mTOR and S6K1, which was followed by a decreased mRNA level of the *CEBPA* gene [267].

Lluch et al. [268] assessed the therapeutic blockage of S6K1 in HFD-induced obese mice challenged with a specific oral S6K1 inhibitor (LY2584702 tosylate), resulting in diminished S6K1 activity that hampered fat mass expansion. These results confirm the potential of reducing adipogenic commitment and adipocyte hypertrophy in adipocyte progenitors, thus inhibiting the onset of obesity by a compound targeting S6K1 in AT [141,268]. Notably, the processing of SREBFs has been found to require S6K1 downstream from mTORC1 and is therefore sensitive to rapamycin [269,270,271].

Treatment of MSCs with eudesmin disturbs adipogenesis via suppression of the S6K1 signaling pathway. Eudesmin treatment inhibits the activation and nuclear translocation of S6K1. Consequently, S6K1-mediated phosphorylation of H2B at serine 36 (H2BS36p) is reduced upon eudesmin treatment, further inducing the expression of *WNT6*, *WNT10A*, and *WNT10B*, which disturbs adipogenic differentiation [272].

Therefore, accumulated evidence links early AA abundance to increased FTO-mTORC1-S6K1 signaling that, via *EZH2*-H3K27me3-mediated *WNT* suppression, promotes postnatal ASC proliferation, thereby increasing the cell number of ASCs (Figure 2).

In contrast, inhibition of this pro-adipogenic signaling cascade at various levels disturbs ASC commitment, adipocyte progenitor cell expansion and adipogenesis (Table 1)

Critical for driving adipogenic commitment is the suppression of *WNT* genes (*WNT6*, *WNT10A*, *WNT10B*), which trigger the upregulation of transcription factors PPARγ and C/EBPα [185,248,273]. Furthermore, PPARγ directly interacts with β-catenin [274]. Yi et al. [258] confirmed that depletion of S6K1 during commitment reduced *PPARG* and *CEBPA* mRNA levels.

Thus, FF-induced overactivation of *FTO*/mTORC1/S6K1 signaling by enhancing nuclear S6K1/*EZH2*-mediated *WNT* gene suppression enhances the expression of adipogenic transcription factors.

#### 2.4.2. The Interplay of Wnt/β-Catenin, PPARγ, and *FTO* in MSC Lineage Commitment

A functional negative interaction between β-catenin and PPARγ is well accepted [275,276], and MSC commitment to either adipogenesis or osteogenesis is well documented [277,278]. The inhibition of the Wnt/β-catenin signaling pathway promotes MSC fate decisions towards adipocyte differentiation [252,273,274,275,276]. In particular, activators of Wnt/β-catenin signaling suppress adipogenesis by inhibiting the expression and activity of PPARγ and C/EBPα, whereas reduction of β-catenin expression by its siRNA transfection coincided with a significantly increased expression of PPARγ and C/EBPα [279]. Decreased expression of both the *WNT10B* and *CTNNB1* genes increased the expression of the *CEBPA* and *PPARG* [280]. Of note, C/EBPα and C/EBPδ promote PPARγ2 expression by directly binding to specific sites in the PPARγ2 promoter [281]. Studies with *CEBPA*-deficient mice demonstrated that C/EBPα is required for the differentiation of WAT, but not BAT [282]. Intriguingly, both adipogenic transcription factors C/EBPα and PPARγ have been shown to promote the expression of *FTO* [173,187].

In fact, *FTO* is upregulated in bone marrow during aging or osteoporosis in humans and mice in a GDF11 (growth differentiation factor 11)-C/EBPα-dependent mechanism [173]. The expression of *FTO* was also upregulated during adipogenic differentiation of BMSCs, whereas its expression was downregulated during osteoblast differentiation. Gain-of-function and loss-of-function experiments showed that *FTO* favored the BMSCs to differentiate to adipocytes rather than osteoblasts. Notably, *FTO* demethylated the *PPARG* mRNA, increasing its expression [173]. Furthermore, Chen et al. [283] showed that overexpression of *FTO* promotes adipogenesis through inhibition of Wnt/β-catenin signaling via suppression of β-catenin in porcine intramuscular preadipocytes. β-Catenin, after translocation into the nucleus and binding to the LEF/TCF family, suppresses *CEBPA* and *PPARG* expression [283]. In contrast, *FTO* silencing decreased the level of phospho-histone H3 protein and inhibited the proliferation of porcine intramuscular preadipocytes, downregulated the expression of PPARγ and C/EBPα but upregulated the expression of β-catenin [283].

Adipose-specific PPARγ knockout mice display abnormalities in the formation and function of both WAT and BAT and show diminished weight gain when fed an HFD [284]. Thus, a complex molecular crosstalk exists between Wnt/β-catenin signaling and the expression of C/EBPα, PPARγ, and *FTO*. Suppressed Wnt/β-catenin signaling but enhanced C/EBPα, PPARγ, and *FTO* expression drive both late steps of adipogenesis and early steps of ASC determination (Figure 3).

#### 2.4.3. FTO-Mediated BMP2- and BMP4-Signaling

Yi et al. [258] tracked the cellular location of S6K1 in MEF-derived 10T1/2 cells and observed that cytoplasmic S6K1 rapidly translocated into the nucleus upon exposure to the adipogenic inducer bone morphogenetic protein 4 (BMP4), which enhanced chromatin recruitment of S6K1, and increased H2BS36p and H3K27me3 at *WNT* promoters. Remarkably, *FTO* upregulates the expression of *BMP4* in an m^6^A-dependent manner and binds to the N-terminal of BMP4 to form a dimer at the C-terminal in cervical cancer cells through protein–protein interaction [285]. BMP4 is a member of the transforming growth factor-β (TGF-β) superfamily of cytokines responsible for stem cell commitment to differentiation, proliferation, and maturation. The adipocyte commitment process can be initiated by either BMP2 or BMP4 [286,287]. In particular, BMP4 is capable of triggering commitment of pluripotent C3H10T1/2 stem cells to the adipocyte lineage [286]. In fact, BMP4 has been implicated in the commitment of pluripotent MSCs to the adipocyte lineage by two independent lines of investigation. First, growth-arrested 10T1/2 cells do not normally respond to a hormonal cocktail that causes various growth-arrested preadipocyte cell lines to differentiate into adipocytes, but if 10T1/2 cells are first treated with BMP4, they will respond to these hormonal inducers by undergoing terminal adipocyte differentiation [287]. Second, a preadipocyte cell line, A33 cells, derived from 10T1/2 cells after exposing the cells to the DNA methyltransferase inhibitor 5-azacytidine, was shown to express BMP4, and this endogenous BMP4 expression is required for acquisition of the preadipocyte phenotype of these cells [287]. BMP4 proved to be one of the popular differentiation factors to induce ASC differentiation into cells of mesodermal origin [288,289].

Of note, the expression of *BMP2* also depends on m^6^A demethylation and is regulated by the m^6^A demethylase ALKBH5 through activated AKT signaling in ligamentum flavum cells [290]. It has recently been shown in suture MSCs that *FTO* knockdown or *FTO* inhibition by a small molecule inhibitor of *FTO* reduced the expression of BMP2 [291].

Taken together, the AA-sensing and AA-inducible m^6^A demethylase *FTO* exerts downstream regulatory effects on both, the expression level of Wnt and BMP proteins, crucial mediators of MSCs, which promote adipocyte lineage commitment, the first key step of adipogenesis [286,292].

#### 2.4.4. Amino Acid/S6K1-Mediated Inhibition of GSK3 Impairs *FTO* Degradation

The α- and β-isoform of glycogen synthase kinase 3 (GSK3) are both inhibited by S6K1-mediated inhibitory phosphorylation [293,294]. AA abundance indirectly inhibits the activity of GSK3 [294]. Increasing the concentration of AAs in culture media via activation of S6K1 enhances the inhibitory phosphorylation of both GSK3 isoforms [295]. Furthermore, it has been demonstrated in L6 muscle cells that L-leucine availability upregulates S6K1 while inactivating GSK3β [296]. Remarkably, Faulds et al. [297] reported that GSK3 activity regulates m^6^A mRNA methylation in mouse embryonic stem cells (MECs). In particular, GSK3-mediated phosphorylation of *FTO* leads to *FTO* polyubiquitination and proteasomal degradation. GSK3 knockout in ESCs thus results in elevated *FTO* protein levels and 50% less m^6^A than WT ESCs, pointing to a putative novel mechanism by which GSK3 activity regulates stem cell pluripotency [297].

GSK3 is a multifaceted kinase also critically involved in Wnt signaling [298]. Three pools of GSK3 apparently exist. One pool of GSK3 is associated with AXIN and regulated by low-density lipoprotein receptor-related protein 5/6 (LRP5/6). Another pool depends on phosphorylation by the PI3K-AKT pathway and appears to be an AXIN-independent pool of GSK3 that might also be regulated by Wnt [298]. Bennett et al. [252] demonstrated that a specific GSK3 inhibitor (CHIR 99021) mimics Wnt signaling in preadipocytes, stabilizing free cytosolic β-catenin and inhibiting adipogenesis by blocking the induction of C/EBPα and PPARγ. Preadipocyte differentiation is inhibited when 3T3-L1 cells are exposed to CHIR99021 during the first 3 days of adipogenesis. Zaragosi et al. [299] reported that GSK3 inhibitors suppressed in vitro expansion and differentiation of human ASCs into adipocytes.

Further studies are required to investigate differential effects of AA-S6K1-mediated GSK3 inhibition by intake of high- versus low-protein FF on GSK3-FTO- and GSK3-Wnt signaling on ASC fate decisions.

#### 2.4.5. Ciliary FTO- and Wnt Signaling Related to ASC Commitment

Primary cilia are highly conserved membrane-associated hair-like organelles that are important for cellular signaling [300,301,302,303]. The cilium is considered to function as a cellular antenna that influences obesity risk [304]. The life cycle of a primary cilium begins in quiescence and ends prior to mitosis [305]. Of note, primary cilium decapitation induces mitogenic signaling of cells and drives the cell cycle, supporting the role of ciliary functions in cell proliferation [306]. Current evidence suggests that primary cilia coordinate various signaling pathways, including those regulated by Wnt, TGFβ/BMP, hedgehog (HH), G protein-coupled receptors (GPCRs), and receptor tyrosine kinases (RTKs), to control developmental processes, tissue plasticity, and organ function [306]. Thus, primary cilia are involved in regulating key mediators that promote MSC commitment to preadipocytes, while impaired cilia compromise ASC functions [307].

Syndromic forms of obesity have been identified as ciliopathies, including Bardet–Biedl syndrome and Alström syndrome [308]. The primary cilium is critically involved in multiple ciliary signaling pathways, involving Wnt, HH and RTK signaling, which regulate adipogenic differentiation [303,309]. Notably, the differentiation of MSCs into adipocytes requires coordination of external stimuli and depends on the functionality of the primary cilium [310]. Rab8, a member of the Rab family of small GTPases, is localized in the primary cilium and may direct vesicle docking and fusion to the cilium base [310]. Rab8 has been shown to attenuate Wnt signaling and is required for mesenchymal differentiation into adipocytes [310]. Primary cilia assembly and function are also necessary to upregulate the expression of pro-adipogenic factors CEBPs and PPARγ during adipogenesis [311,312,313,314]. Impaired primary cilia may prevent preadipocytes from differentiation, whereas existing mature adipocytes display increased fat accumulation [311,314]. Furthermore, the primary cilium may also function in inhibiting Wnt signaling by promoting β-catenin degradation [303,315,316]. Notably, the cilium, through regulated intraflagellar transport (IFT), diverts Jouberin (Jbn), a ciliopathy protein and specific Wnt pathway regulator, away from the nucleus and limits β-catenin nuclear entry [316].

Intriguingly, *FTO* plays a critical regulatory role for the primary cilium and its signaling function. In zebrafish, loss of *FTO* results in short, absent, or disorganized cilia [317]. Loss of retinitis pigmentosa GTPase regulator interacting protein 1-like (RPGRIP1L) in 3T3-L1 preadipocytes increased the number of cells that are capable to differentiate into mature adipocytes [318]. *RPGRIP1L* is an evolutionarily highly conserved gene encoding a protein that localizes at the transition zone of primary cilia [319]. Mutations in *RPGRIP1L* result in ciliopathies [319].

The CUX1 regulatory element within the implicated obesogenic *FTO* region controls the expression of *FTO* and the nearby ciliary gene, *RPGRIP1L* [320]. Hypomorphism for RPGRIP1L, a ciliary gene adjacent to the *FTO* locus, causes increased adiposity in mice [320]. CUX1, belonging to the CpG sites exhibiting significant DNA methylation variations associated with exclusive BF and longitudinal BMI [321], is suggested to control the body mass of mice and humans by influencing RPGRIP1L [322]. However, the expression of RPGRIP1L is unaltered in FTO-negative mice, arguing against this hypothesis [91]. In RPGRIP1L-negative MEFs, decreased autophagic activity was due to increased ciliary activity of mTORC1 [323]. In confluent 3T3-L1 preadipocytes, the primary cilium and its basal body form an organized signaling pathway for the IGF-1 receptor to induce adipocyte differentiation [324].

A recent study has revealed a conserved role of the *FTO*–FOXJ1 axis in embryonic and homeostatic motile ciliogenesis [325]. Remarkably, *FTO* demethylates and thereby stabilizes the mRNA that encodes the master ciliary transcription factor FOXJ1 [325]. FOXJ1 functions in the postcentriologenesis stage by establishing mechanisms for docking of basal bodies at the apical membrane and induction of programs of axoneme assembly [326]. FOXJ1 regulates floor plate cilia architecture and modifies the response of cells to sonic HH signaling [327]. The primary cilium undergoes dynamic size modifications during adipocyte differentiation of human ASCs [328]. Of note, HH signaling, an anti-adipogenic pathway dependent on the primary cilium, is inhibited after three days of differentiation, concomitantly with the cilium size increase [328]. Vertebrate HH signaling requires the primary cilium for multiple aspects of signal transduction [329]. As shown in the multipotent MSCs (C3H10T1/2 cells), the addition of sonic HH to the medium inhibited adipogenic differentiation [330]. It has also been shown in muscle-resident fibro/adipogenic progenitors that ciliary HH signaling restricts adipogenesis [331].

Taken together, accumulated evidence links increased *FTO* signaling in MSCs/ASCs to downstream ciliary signaling pathways including Wnt, BMP2, BMP4, and sonic HH, the key mediators regulating adipocyte lineage commitment of MSCs.

#### 2.4.6. Impact of Amino Acid Availability on Stem Cell Proliferation and Renewal

Recent interest focuses on the influence of AA-mediated metabolism in stem cell homeostasis and function [332,333], because the self-renewal and differentiation potentials of stem cells are dependent on AA metabolism [332]. Sartori et al. [334] demonstrated that supplementation with BCAA led not only to increased MSC proliferation with more cells in the S, G2, and M cycle phases, but also to increased metabolic activity. Intriguingly, BCAA supplementation of MSCs increased the expression of PPARγ [334], the key transcription factor essential for adipocyte differentiation, maintenance, and function [335]. In immortalized fibroblasts lacking PPARγ, C/EBPα alone was not able to promote adipogenesis, indicating that PPARγ is the proximal effector of adipogenesis [336]. Two potential antagonists of PPARγ (i.e., BADGE and GW9662), as well as lentivirus-mediated knockdown of PPARγ, inhibited hMSC adipogenesis [337]. PPARγ is critical for promoting MSC adipogenic differentiation and adipocyte lineage commitment, as well as the reciprocal regulation of adipogenesis and osteogenesis [338,339,340,341,342]. Importantly, Jiang et al. [343] recently showed that the assembly and maintenance of the adipose vascular niche are controlled by PPARγ acting within APCs. PPARγ triggers a molecular hierarchy that induces vascular sprouting, APC vessel niche affinity, and APC vessel occupancy. Mechanistically, PPARγ transcriptionally activates PDGFRβ and vascular endothelial growth factor (VEGF). APC expression and activation of PDGFRβ promote the recruitment and retention of APCs to the niche [343]. Interestingly, PPARγ upregulates the expression of L-type AA transporters (LAT1 and LAT2) and taurine transporter (TAU), which was reversed by PPARγ siRNA [344]. AA uptake via TAU has been shown to regulate adipogenic differentiation of human ASCs through affecting the Wnt/β-catenin signaling pathway [345].

Remarkably, adipocyte deletion of one PPARγ copy increased mouse serum BCAA and reduced inguinal WAT and BAT adipose tissue BCAA incorporation into triacylglycerol, as well as mRNA levels of branched-chain aminotransferase 2 (BCAT2) and branched-chain α-ketoacid dehydrogenase (BCKDH) complex subunits. Adipocyte deletion of two PPARγ copies induced lipodystrophy, severe glucose intolerance, and markedly increased serum BCAA levels [346]. Increased intracellular uptake of BCAAs is a requirement for the activation of mTORC1 [113,114,115]. mTORC1 participates in the lineage determination of MSCs and promotes adipogenesis in white adipocytes, brown adipocytes, and muscle satellite cells, while rapamycin inhibits the adipogenic function of mTORC1 [347]. Notably, mTORC1 activation supports the translation of PPARγ [141] and plays a critical role in 3T3-L1 preadipocyte differentiation and requiring its kinase activity [348]. The expression of S6K1, a well-known downstream target of mTORC1 kinase, increases during adipogenesis [348]. Importantly, PPARγ activity has been shown to require AA sufficiency dependent on mTORC1-mediated nutrient-sensing [348].

Thus, increased insulin-, IGF-1-, and BCAA serum levels of formula-fed infants may overstimulate mTORC1-, S6K1-, and PPARγ-dependent adipocyte progenitor cell expansion (Figure 4).

*PPARG* is a direct target of miR-130a-3p [349,350], which regulates the biosynthesis of bovine milk fat by targeting *PPARG* [349] and is a component of HBM EVs [177]. Thus, miR-deficient infant formula lacks the postnatal regulatory capacity to adjust milk miR-mediated *PPARG* expression, potentially enhancing PPARγ-mediated adipocyte progenitor cell expansion.

The AA transporter ASC-1 is a white adipocyte-specific cell surface protein, with little or no expression in brown adipocytes [351,352]. The AA transporter ASC1/CD98hc (SLC7A10/SLC3A2) is a non-stereoselective small neutral AA exchanger (e.g., glycine, L- and D-serine, alanine, cysteine, and threonine) expressed mainly in the brain and adipose tissue [353]. Suwandhi et al. [354] identified a subset of adolescent murine preadipocytes expressing the mature white adipocyte marker ASC-1 that showed a low ability to differentiate into beige adipocytes compared to ASC-1 negative cells in vitro. Loss of ASC-1 in subcutaneous preadipocytes resulted in spontaneous differentiation of beige adipocytes in vitro and in vivo. Mechanistically, this was mediated by a function of the AA transporter ASC-1 specifically in proliferating preadipocytes involving the intracellular accumulation of the ASC-1 cargo D-serine [353]. In contrast, Arianti et al. [355] found that ASC-1 transporter-dependent AA uptake is required for the efficient thermogenic response of human beige/brown adipocytes to adrenergic stimulation. Noteworthy to mention is that *SLC7A10* is a target of miR-30-5p [356]. It has been shown in human BMSCs that the osteogenic transcription factor RUNX2 is also a downstream target of miR-30a-5p [357]. In contrast, miR-30a-5p regulates 3T3-L1 cell differentiation by targeting *SIRT1*, thus accelerating adipogenesis by negatively regulating sirtuin 1 [358]. The abundance of miR-30b-5p in HBMEs of obese mothers was 42% lower compared to miR-30b-5p levels in HBMEs of mothers with normal weight [163]. Notably, miR-30b-5p was positively associated with infant weight, percent body fat, and fat mass at 1 month [163].

Armstrong et al. [359] observed in the Drosophila model that a reduction in AA levels, which is associated with an increase in uncharged tRNAs and activation of the general control nonderepressible 2 (GCN2)-dependent AA sensing signaling pathway within adipocytes, increased the rates of germline stem cell (GSC) loss. Under high AA levels, the AA response (AAR) pathway is off and TORC1 is active, resulting in optimal GSC maintenance and ovulation rates. Under lower AA levels, the AAR pathway is triggered through an increase in unloaded tRNAs and activation of GCN2 kinase, leading to GSC loss [359].

According to Kilberg et al. [360], nutrient availability has a strong influence on stem cell growth, self-renewal, and lineage specification, both in vivo and in vitro. Evidence from several laboratories has documented that self-renewal and differentiation of mouse ESCs are critically dependent on proline metabolism, with downstream metabolites possibly serving as signal molecules. Likewise, catabolism of either threonine (mouse) or methionine (human) is required for growth and differentiation of ESCs because these AAs serve as precursors for donor molecules used in histone methylation and acetylation [360].

There is also a close relationship between cancer stem cells (CSCs) and AA availability and metabolism particularly for self-renewal, survival, and stemness recently reviewed elsewhere [361]. Notably, leukemia stem cells (LSCs) isolated from de novo acute myeloid leukemia (AML) patients are uniquely reliant on AA metabolism for oxidative phosphorylation and survival [362]. Pharmacological inhibition of AA metabolism reduces oxidative phosphorylation and induces cell death [362]. The majority of AAs are able to form tricarboxylic acid (TCA) intermediates [363]. The main function of the TCA cycle is to support NADH generation for oxidative phosphorylation (OXPHOS). It has recently been shown that direct stimulation of NADP^+^ synthesis is mediated through growth factor signaling via AKT-mediated phosphorylation of NAD kinase [364]. Importantly, NADP via enhancing *FTO* activity reduces RNA m^6^A methylation and increases adipogenesis [155].

AA abundance is thus linked to OXPHOS and NADP-stimulated activation of *FTO*.

#### 2.4.7. Impact of Amino Acids on Oxidative Phosphorylation During ASC Differentiation

Marked mitochondrial biogenesis and OXPHOS occur during preadipocyte and adipocyte differentiation [365,366,367,368]. Changes in protein expression include enzymes and transporters involved in the TCA cycle, fatty acid oxidation, and ATP synthesis [365]. The identified proteins included enzymes and transporters involved in the TCA cycle, fatty acid oxidation, and ATP synthesis [365]. 3T3-L1 preadipocyte differentiation was associated with a 20- to 30-fold increase in the concentration of numerous mitochondrial proteins [366]. The protein expression of cytochrome c and the enzyme activity of cytochrome c oxidase (COX) increased with porcine preadipocyte differentiation [367]. Both nondifferentiated and differentiated 3T3-L1 adipocytes meet ATP demand primarily through oxidative phosphorylation [368].

##### GADD45G-Interacting Protein

Mitochondrial oxidative phosphorylation reserve is required for hormone- and PPARγ agonist-induced adipogenesis [369]. To investigate the role of OXPHOS in adipogenesis, Ryu et al. [369] analyzed adipocyte differentiation following disruption of CRIF1 in vitro and in vivo. CRIF1, also known as growth arrest and DNA-damage-inducible protein-interacting protein, is encoded by the *GADD45GIP1* gene. CRIF1 is a translational factor for mitochondrial DNA (mtDNA) and is important for transcription of the mitochondrial OXPHOS complex. Remarkably, the adipose-specific CRIF1-knockout mouse had a lower body weight and less fat mass than wild-type mice. In addition, adipocytes were smaller and exhibited a dysplastic morphology. 3T3-L1 adipocytes or ASCs lacking *CRIF1* expressed lower levels of mtDNA-encoded OXPHOS subunits, and adipocyte differentiation was disrupted [369]. Remarkably, recent evidence indicates that *GADD45GIP1* belongs to a gene network of human β-cells of T2DM significantly co-expressed with m^6^A regulators, including *FTO* [370]. It is likely that AA-induced *FTO* overexpression enhances OXPHOS in ASCs promoting ASC self-renewal and proliferation, resulting in an increased number of ASCs, a potential adverse effect of excessive protein intake during the lactation period. HBMEs deliver abundant miR-30-5p, which targets *GADD45GIP1* [371].

##### Peroxisome Proliferator-Activated Receptor-γ Coactivator 1β

The expression of peroxisome proliferator-activated receptor-γ coactivator 1β (PGC-1β) encoded on the *PPARGC1B* gene improves insulin sensitivity and mitochondrial function in 3T3-L1 adipocytes [372]. Overexpression of PGC-1β in 3T3-L1 preadipocytes showed a broadening of mitochondrial cristae, an increase in mitochondrial DNA and fission 1 protein (Fis1) mRNA expression, and increased intracellular ATP levels [372]. PPARγ and PGC-1β are coordinately upregulated in adipocytes relative to preadipocytes. Thiazolidinedione treatment induces PGC-1β and mitochondrial marker genes in the AT of obese mice [373]. Merkestein et al. [96] observed that *FTO* overexpression in FTO-4 mice resulted in the upregulation of PGC-1β. *PPARGC1B* has been identified as a m^6^A-regulated gene [374]. Overexpression of PGC-1β upregulates the expressions of adipogenic and mitochondrial biosynthetic marker genes and promotes triglyceride accumulation during 3T3-L1 adipocyte differentiation, whereas siRNA silencing of *PPARGC1B* inhibited the expressions of mitochondrial genes, PPAR-γ, SREBF1c, and fatty acid synthetase, resulting in a reduction of triglyceride synthesis [375]. Of note, *PPARGC1B* mRNA is targeted by the conserved let-7 family of miRs (let7a, let7b, let7c, let7-d, let7g, let7i, let7e, let7f) [376,377]. Let-7 family members belong to the highly conserved miRs in milk, milk fat, skim milk, and milk exosomes [170,182,183,378,379,380,381,382]. In fact, miR-148a-3p, miR-30a/d-5p, miR-22-3p, miR-146b-5p, miR-200a/c-3p, and the 5p end of the let-7 miRs were commonly reported among the top 10 miRs in the cell, lipid, and skim milk fractions of HBM [378,382]. Abundance of circFUT10 via sponging let-7c has been shown to promote adipocyte proliferation and inhibit cell differentiation in bovine fat tissue [377]. Thus, HBME-derived let-7 miRs may counteract adipocyte proliferation via targeting *PPARGC1B*, the transcriptional regulator of mitochondrial function and OXPHOS. In contrast, miR-deficient formula apparently fails in early postnatal miR-dependent regulation of ASC proliferation.

##### Estrogen-Related Receptors

Orphan members of the superfamily of nuclear estrogen-related receptors (ERRs) are crucial for controlling mitochondrial gene networks in adipocytes of WAT and BAT [383,384,385]. ERRα encoded on the *ESRRA* gene and ERRα-related transcriptional coactivators PGC-1α and PGC-1β can be upregulated in 3T3-L1 preadipocytes at the mRNA level under adipogenic differentiation conditions, including inducers like cAMP, glucocorticoids, and insulin [385]. Inhibiting *ESRRA* with specific siRNA leads to the downregulation of mRNA for of fatty acid binding protein 4 (FABP4), PPARγ, and PGC-1α, in 3T3-L1 cells in adipogenesis medium. ERRα and PGC-1β mRNA expression can also be upregulated in other preadipocyte lineages, such as DFAT-D1 cells and the pluripotent mesenchymal cell line C3H10T1/2, under differentiation conditions. Furthermore, stable expression of ERRα in 3T3-L1 cells upregulates adipogenic marker genes and promotes triglyceride accumulation during differentiation [386].

Notably, the concentration of miR-148a-3p and miR-125b-5p in HBM decreases in transitional milk around day 30 postpartum [160]. *ESRRA* mRNA is a predicted target gene of miR-125b-5p [387]. Thus, HBM-derived miR-125b-5p may control the transcriptional network of ERRα-dependent regulation of ASCs and early adipocyte progenitors. In accordance, pharmacological inhibition of *ESRRA* protects obese mice against bone loss and high marrow adiposity [387].

Kubo et al. [388] reported that ERRγ encoded on the *ESRRG* gene is upregulated in murine mesenchyme-derived cells, especially in ST2 and C3H10T1/2 cells, at the mRNA level under adipogenic differentiation conditions, including inducers like cAMP, glucocorticoids, and insulin. In accordance, knockdown of *ESRRG* with specific siRNA results in the downregulation of adipogenic marker genes, including FABP4, PPARγ, and PGC-1β, in media of 3T3-L1, ST2, and C3H10T1/2 cell lines. Conversely, ERRγ expression positively regulates adipocyte differentiation in 3T3-L1 cells, leading to the upregulation of adipogenic marker genes [388].

Interestingly, *ESRRG* mRNA is targeted by highly conserved miRs found in HBM, including miR-148a-3p, miR-200b/c-3p, miR-30-5p, and miR-125-5p [389]. *ESRRG* expression and splicing are also epigenetically controlled by m^6^A methylation [390].

#### 2.4.8. Insulin- and IGF-1-Signaling Stimulate ASC Proliferation

Following commitment to the adipose lineage, growth-arrested preadipocytes can differentiate into adipocytes by treatment with IGF-1, glucocorticoid, and an agent that increases cAMP level associated with a rapid and transient increase in C/EBPβ and synchronous re-entry into the cell cycle [391]. Miki et al. [392] studied the contribution of IRS-1 and IRS-2 in MEFs on adipocyte differentiation. The abilities of IRS-1^−/−^ cells and IRS-2^−/−^ cells to differentiate into adipocytes are approximately 60% and 15% lower, respectively, than that of wild-type cells, at day 8 after induction. Double-deficient IRS-1^−/−^/IRS-2^−/−^ cells have no ability to differentiate into adipocytes [392]. IRS-1^−/−^/IRS-2^−/−^ cells exhibit significantly decreased mRNA and protein expression of C/EBPα and PPARγ [392]. PI3K activity, which increases during adipocyte differentiation, is almost completely abolished in IRS-1^−/−^/IRS-2^−/−^ cells [392]. Furthermore, IRS-1^−/−^/IRS-2^−/−^ double-knockout mice 8 h after birth reveal a severe reduction in WAT mass [392]. It is important to note that *IRS1* mRNA is targeted by miR-30-5p and miR-200bc-3p [393], whereas *IRS2* mRNA is a target of the conserved let-7-5p family (let-7a, let7b, let7c, let7d, let7e, let7f, let7g, let7i) as well as miR-30-5p [394], which are all miRs delivered by BF.

Of note, PI3K inhibition (with LY294002) of wild-type cells reduced the expression of C/EBPα and PPARγ, completely inhibiting adipocyte differentiation [392]. Janderová et al. [395] showed that noncommitted precursors of human MSCs are able to differentiate into mature adipocytes after exposure to insulin, dexamethasone, indomethacin, and 3-isobutyl-1-methylxanthine three times for 3 days each. In accordance, active insulin signaling increases the pro-adipogenic potential of BMSCs, leading to bone marrow adipose tissue expansion associated with enhanced insulin sensitivity, glucose uptake, and OXPHOS [396].

Scavo et al. [397] evaluated the role of the IGF-1 receptor (IGF1R) in the process of adipocyte differentiation in bone marrow-derived human MSCs and demonstrated that nanomolar concentrations of IGF-1 adequately replaced micromolar concentrations of insulin in supporting differentiation and lipid accumulation in human MSCs. In fact, the addition of IGF-1 specifically increased cell proliferation and lipid accumulation in human MSCs [397]. IGF-1 also increased the expression of PPARγ, whereas this effect was blocked by treatment of human MSCs with an αIR-3 antibody, which inhibits IGF1R activity [397]. Hu et al. [398] provided further evidence showing that IGF-1 promotes adipogenesis by a lineage bias of endogenous adipose stem/progenitor cells. In particular, IGF-1 attenuated Wnt/β-catenin signaling by activating Axin2/PPARγ pathways in SVF cells, suggesting that IGF-1 promotes CD31-/34^+^/146^−^ bias through tuning Wnt signaling.

It may not be accidental, that *IGF1R* mRNA is targeted by the conserved let-7-5p family (let-7a, let-7b, let-7c, let-7d, let-7e, let-7f, let-7g, let-7i) and miR-30-5p [399], key miRs of HBM and HBMEs.

Thus, accumulated evidence underlines the role of insulin/IGF-1/IGF1R/IRS-1/IRS-2/PI3K/AKT signaling cascade in ASC commitment and differentiation.

#### 2.4.9. Suppression of TP53 Enhances ASC Proliferation

Increased insulin/IGF-1 signaling activates the kinase AKT, which inhibits the activity of the transcription factor p53 [400]. In particular, AKT activated by exogenous IGF-1 promotes the phosphorylation of the p53-binding protein mouse double minute 2 (MDM2), which increases the ability of MDM2 to degrade p53 [400,401]. Thereby, MDM2 promotes cell survival and cell cycle progression by inhibiting p53. To regulate p53, MDM2 must gain nuclear entry, thereby diminishing cellular levels of p53 and decreasing p53’s transcriptional activity [401]. Phosphorylated MDM2 finally promotes the proteasomal degradation of p53 [402,403,404]. Thus, IGF-1/AKT signaling lowers the level of nuclear p53, which may enhance ASC differentiation [405]. Notably, Berberich et al. [406] discovered that 3T3-L1 cells possess a 36-fold elevation of MDM2 mRNA relative to A31 cells, another immortalized Balb/c 3T3 fibroblast cell line that lacks the capacity to differentiate. During the conversion of undifferentiated fibroblasts to adipocytes, MDM2 mRNA levels remained elevated, whereas p53 mRNA, protein, and DNA-binding activity decreased [406]. Although these results suggest that p53 is a negative regulator of the adipogenic program, data about p53 signaling in 3T3-L1 cells need to be interpreted with caution [407], since in the transformed mouse 3T3 cells, the MDM2 gene is highly amplified [406]. Basal p53 expression of p53 is indispensable for MSC integrity [408].

Nevertheless, p53 plays a crucial role in white adipocyte differentiation and function [406,407]. The multipotent C3H10T1/2 mouse cell line, capable of differentiating into white adipocytes, is valuable for studying early adipogenesis events [407]. Knockdown of p53 in C3H10T1/2 cells resulted in enhanced differentiation with increased expression of adipogenic genes throughout the process [409]. Through various in vitro models representing different stages of white adipocyte differentiation, it has been demonstrated that p53 suppresses white adipocyte differentiation in both mouse and human cells [409]. Upon adipogenic induction of the multipotent mouse stromal MBA-15 cell line, p53 knockdown led to elevated levels of PPARγ and increased adipocyte colony formation [410]. Similar results were observed in human ASCs [407]. Furthermore, MEFs isolated from p53 knockout mice were able to differentiate into adipocytes [410,411]. Okita et al. [411] found that p53 stabilization or overexpression in 3T3-L1 preadipocytes downregulates the expression of *PPARGC1A*, the master regulator of mitochondrial biogenesis in 3T3-L1 preadipocytes and MEFs. Conversely, p53 downregulation enhanced differentiation into adipocytes and mitochondrial DNA content. Furthermore, p53-depleted 3T3-L1 cells exhibited increased mitochondrial proteins and enhanced the activities of citrate synthase and complex IV during adipogenesis [411]. In fact, p53-knockout MEFs spontaneously committed to the adipogenic program, while ectopic re-expression of p53 efficiently inhibited their spontaneous adipogenic differentiation [410,412]. Furthermore, nutlin-3a-mediated p53 accumulation downregulated PPARγ in wild-type MEFs [410]. This negative role of p53 in adipocyte differentiation in cell lines and MEFs is supported by data from primary cells derived from the SVF of inguinal WAT or interscapular BAT from p53 knockout mice [413]. TP53 inhibits the activity of PPARGC-1α and PPARGC-1β [413]. Knockdown of p53 in human ASCs increased the differentiation potential [409]. Hallenborg et al. [413] suggested that the ability of p53 to inhibit adipogenesis depends on its DNA binding, as mutations to the DNA-binding domain of p53 failed to inhibit adipocyte conversion of wild-type MEFs. Collectively, these results indicate an inhibitory function of p53 in adipogenesis.

MEFs deficient in the cyclin-dependent kinase inhibitor 1a (CDKN1A; p21), which is upregulated by p53 [414,415], underwent spontaneous adipose conversion [412]. In accordance, knockout of p21 in mice induced adipocyte hyperplasia [416]. p53 is a negative regulator of *IGF1R* [417,418] and mTORC1/S6K1 signaling [419,420,421]. In fact, phosphorylated S6 (pS6) is increased in the hearts of p53-deficient mice, correlating with increased body weight [421]. In addition, p53^−/−^ mice showed slight hyperinsulinemia with elevated IGF-1 levels [421]. Insulin/IGF-1AKT-induced suppression of p53 signaling may thus enhance mTORC1/S6K1 signaling with subsequent nuclear *EZH2*-mediated suppression of *WNT* gene expression, crucial for early adipocyte progenitor differentiation.

A dynamic crosstalk between p53 and Wnt has been reported [422,423,424]. Wnt/β-catenin signaling regulated the proliferation and differentiation of mesenchymal progenitor cells (MPCs) through the p53 pathway [423]. Wnt/β-catenin signaling positively regulated p53 expression. Silencing of p53 increased the proliferation and differentiation of MPCs [424]. Conversely, induction of Wnt ligand genes by p53 has been associated with an anti-differentiation role of p53 in murine ESCs. Notably, Wnt signaling pathway genes, especially WNT8B, were significantly induced by adriamycin treatment in p53^+/+^ murine ESCs compared to p53^−/−^ murine ESCs [424].

It is conceivable that FF-induced upregulation of insulin and IGF-1 serum levels affects ASC differentiation in the SVF. Enhanced insulin/IGF-1/AKT via MDM2 phosphorylation and reduced ASC p53 levels, suppressing Wnt signaling area critical steps for early adipocyte progenitor cell development [249,253]. Furthermore, it has been shown in human kidney proximal tubular cells (HK2 cells) that *FTO* overexpression reduced p53 expression, whereas meclofenamic acid-mediated inhibition of *FTO* upregulated p53 mRNA and protein levels [425]. In accordance, *FTO* knockdown by siRNA promoted p53 mRNA by 20.4-fold at 24 h after cisplatin administration in HK2 cells [425]. Recent evidence indicates that p53 mRNA exhibits m^6^A modifications that are of functional importance [426]. Zhao et al. [427] demonstrated in arsenite-transformed human keratinocytes that a high m^6^A level inactivated p53 by modulating the expressions of p53 regulators. However, chronic arsenic exposure increases the expression of m^6^A methyltransferases but decreases the expression of *FTO* [427], a constellation that may not correspond to FF-induced overexpression of *FTO* [110]. Noteworthy to mention is that C/EBPβ, a transcription factor upregulated via FTO-mediated m^6^A-demethylation [130], suppresses the expression of p53 [428,429], a critical interaction enhancing MCE of preadipocytes [227].

Nevertheless, further experimental studies are required to clarify the role of FF and interactions between *FTO*, p53, and Wnt in regulating early ASC homeostasis.

### 2.5. Potential Impact of Milk Exosomal MicroRNAs on the Adipocyte Stem Cell Niche

A recent review suggests that exosomes from various origins and body fluids influence the adipogenic differentiation of stem cells [430]. MiRs delivered by exosomes and EVs are key regulators of stem cell processes and functions [431,432,433] and are critically involved in MSC fate decisions balancing adipogenesis and osteogenesis [434,435,436]. Several miRs have been identified to play an important role for ASC proliferation, promotion, or inhibition of ASC differentiation [436,437,438,439,440]. A recent miR-gene interaction network of adipogenesis of MSCs discovered the top 10 pivotal miRs, including miR-27a-3p, let-7b-5p, miR-1-3p, miR-124-3p, miR-155-5p, miR-16-5p, miR-101-3p, miR-21-3p, miR-146a-5p, and miR-148b-3p [441]. Ou-Yang et al. [442] screened differentially expressed miRs of adipogenic differentiation and dedifferentiation of MSCs and identified 10 key miRs: miR-27a-3p, miR-182-5p, miR-7-5p, miR-16-5p, miR-1-3p, miR-155-5p, miR-21-3p, miR-34a-5p, miR-27a-5p and miR-30c-5p; some of them are components of HBM.

Remarkably, exosome abundance and exosomal miR composition in milk change during the stages of lactation [443,444]. Milk collected on 3-8 postnatal days has higher exosome concentrations than mature milk collected in the second month [445]. Furthermore, colostrum has higher miR concentrations than HBM [446]. Colostrum is the milk lactated within the first 7 days after birth, whereas transitional milk is lactated 7–14 days after birth, and mature milk is provided from 2 weeks after birth. As the infant’s intestinal permeability is highest during the first postnatal week of neonatal life, colostrum-derived exosomes and their miRs may have the highest opportunity to reach the blood circulation and SVF, affecting ASC development.

Wu et al. [447] recently compared the miR expression of colostrum versus mature HBM. Whereas both share 715 common miRs, colostrum has 67 colostrum-specific miRs that were not detected in mature HBM. KEGG analysis of human colostrum-specific miR targets exhibit signaling pathways regulating pluripotency of stem cells, Wnt signaling, and PI3K-AKT-mTOR signaling, among others [447]. According to Wu et al. [447], the expression levels of miR-30b-5p, miR-885-5p, miR-29c-3p, miR-511-3p, miR-429, and miR-183-5p in colostrum were significantly lower than those in mature HBM, whereas the expression level of miR-623 in colostrum was five times higher than in mature milk (*p* < 0.01) [447]. Let us have a close look at early lactation-derived milk miRs.

#### 2.5.1. MicroRNA-623

Interestingly, miR-623 targets cyclin D1 (*CCND1*) [448]. Hishida et al. [449] showed that the expression of cyclin D1 and cyclin D3, the other D-type cyclins, is transiently induced early during adipocyte differentiation. Knockdown of cyclin D1, D2, or D3 through RNA interference inhibits the differentiation of 3T3-L1 cells into lipid-laden adipocytes. Moreover, the knockdown of cyclin D1 or D3 significantly inhibits MCE. Defects in the pathways regulating intracellular D-type cyclins result in abnormal initiation of stem cell differentiation in various organ systems [450]. Notably, knockdown by siRNAs against cyclin-dependent kinase 1 (*CDK1*) and *CCND1* inhibits cellular proliferation, but promotes adipogenic differentiation [451]. In pig primary stromal-vascular (S-V) cultures, cyclin D1 is found in freshly isolated S-V cells and continues to be expressed during the first 3 days of adipose cell development, with a significant increase in late development at day 9. Elevated cyclin D1 levels are colocalized with C/EBPα beginning at day 3 and remain colocalized with C/EBPα through day 9. Removing insulin from cultures results in a reduction in differentially elevated levels of cyclin D1 [452]. Notably, m^6^A modification of *CCND1* mRNA oscillates in a cell-cycle-dependent manner. *FTO* depletion upregulates *CCND1* m^6^A modification, thereby accelerating the degradation of *CCND1* mRNA and leading to the impairment of G_1_ progression. m^6^A levels are suppressed during the G_1_ phase and enhanced during other phases [230]. BF-mediated delivery of colostrum-derived miR-623 may thus suppress cyclin D1, reducing MCE of ASCs, whereas FF with increased circulatory levels of insulin [61,78,79,80], overexpression of *FTO* in PBMCs [110], and absence of HBME miRs in formula [159,160] may overstimulate cyclin D1-mediated MCE of ASCs in the VSF.

#### 2.5.2. MicroRNA-22-3p

Yun et al. [453] reported that miR-22-3p is the most abundant miR of exosomes in human colostrum. Thery confirmed that miR-30a-5p, miR-22-3p, and miR-26a are commonly observed in colostrum and mature milk of humans, cows, and caprines. These miRs are highly conserved in the colostrum and mature milk of these species, suggesting their possible importance in neonatal growth of mammals [453]. The top 10 most abundant miRs shared between human colostrum and mature human milk are miR-22-3p, miR-141-3p, miR-181a-5p, miR-148a-3p, miR-26a-5p, miR-30a-5p, let7a-5p, miR-27b-3p, miR-146b-5p, and let7f-5p [453].

Verma et al. [454] reported 12 higher expressed immune-related upregulated exosomal miRs in colostrum compared to mature milk: miR-22-3p, miR-148a-3p, miR-106-5p, miR-24-3p, miR-30a-5p, miR-32-3p, miR-101-3p, miR-27a-3p, miR-125b-5p, miR-200b-3p, miR-141-3p, and miR-583, respectively. Of importance, miR-22-3p targets *FTO* [168] and may thereby reduce cyclin D1-dependent MCE of ASCs of WAT [230]. Intriguingly, miR-22-3p targets MYC-associated factor X (MAX) [455] and MYC-binding protein (MYCBP) [456,457], which both enhance the transcriptional activity and function of MYC, a critical transcription factor promoting ASC MCE [132]. Conversely, miR-22-3p, which is overexpressed in HBMEs of mothers with preterm delivery [169,170] promotes BAT development, enhancing thermogenesis [171], and stimulates intestinal epithelial cell proliferation and maturation [164,165]. EV-encapsulated miR-22-3p from BMSCs promotes osteogenic differentiation via *FTO* inhibition [172]. In accordance, the differentiation capacity of BMSCs into osteoblasts was increased by miR-22-3p overexpression [458]. Huang et al. [459] demonstrated that overexpression of miR-22-3p significantly inhibits adipogenic differentiation of human AT-derived MSCs by repressing histone deacetylase 6 (HDAC6), thereby promoting osteoblast differentiation. HDAC6 was reported to directly interact with the key osteogenic transcription factor RUNX2, functioning as a corepressor of RUNX2 in pre-osteoblasts [460]. Thus, miR-22-3p and miR-149-3p, via targeting *FTO* expression, inhibit adipogenic, but potentiate osteogenic, lineage differentiation [176].

Furthermore, overexpression of miR-22-3p via targeting Kruppel-like factor 6 (KLF6) downregulated fibro/adipogenic progenitors (FAP) markedly prevented FAP adipogenesis [461].

Pericytes, which surround the walls of small blood vessels, particularly capillaries and micro-vessels, maintain vascular integrity, participate in angiogenesis, regulate blood flow, and serve as a reservoir for multipotent stem/progenitor cells in white, brown, beige, and bone marrow adipose tissues [462]. In human AT-derived pericytes, overexpression of T cell lymphoma invasion and metastasis 1 (TIAM1) promotes an adipogenic phenotype, whereas its downregulation amplified osteogenic differentiation [463]. TIAM1 regulates the cellular morphology and differentiation potential of human pericytes, representing a molecular switch between osteogenic and adipogenic cell fates [463]. Of note, the N terminus of TIAM1 can influence Rac signaling specificity in a different way by interacting with spinophilin (SPL). In particular, spinophilin binding promotes the plasma membrane localization of TIAM1 and enhances the ability of TIAM1 to activate p70 S6K [464]. SPL KO mice lost body weight, which was associated with increased expression of browning maker genes in visceral WAT [465]. Intriguingly, miR-22-3p targets *TIAM1* mRNA and reduces TIAM1 protein expression [466,467], whereas SPL (*PPP1R9B*) is a predicted target of miR-148a-3p [468]. Thus, early lactation-derived milk miR-22-3p/miR-148a-3p signaling may attenuate WAT development but promotes WAT browning, a supportive mechanism for postnatal thermogenesis. Of interest, miR-22-3p also targets the Wnt signaling inhibitor secreted frizzled-related protein 2 (*SFRP2*) [469], thus promoting Wnt signaling that inhibits adipogenesis but stimulates osteogenesis [273,276,277].

Early lactation stage-mediated HBME miR-22-3p signaling may thus increase Wnt signaling, suppressing ASC commitment and subsequent adipogenesis.

Kupsco et al. [177] reported a decrease in human milk EV-transported miR-1290, miR-130a-3p, miR-146a-5p, miR-195-5p, miR-27b-3p, miR-34a-5p, miR-612, miR-6799-5p in 30–80 days post-delivery compared to the first week of lactation. It may be rewarding to look deeper into potential relations of these postnatally declining HBME miRs with respect to ASC regulation and adipogenesis.

#### 2.5.3. MicroRNA-1290

MiR-1290, first discovered in human ESCs, plays an essential role in developing the fetal nervous system [470]. EVs derived from AT-derived stromal cells have been shown to stimulate angiogenesis particularly via miR-1290 [471]. There are almost no substantiated data demonstrating an involvement of miR-1290 in ASC regulation. However, predicted target genes of miR-1290 (*FTO*, *BMP4*, *WNT4*, *WNT5A*, *WNT7A*, *WNT7B*, *IGF1*, *IGF1R*, *IGF2R*, *IRS1*, *IRS2*, *INSR*, *AKT3*, *PIK3CA*, *ESRRG*, *RUNX1T1*, *FABP4*, *PPARGC1B*) point to this direction [472]. BF-derived miR-1290 may thus counteract obesiogenic signaling of FF (upregulation of insulin, IGF-1, BCAAs, and *FTO*).

#### 2.5.4. MicroRNA-146a-5p

Wu et al. [473] demonstrated in primary porcine adipocytes that miR-146-5p inhibits adipogenesis by attenuating insulin receptor (*INSR*) expression and reducing tyrosine phosphorylation of IRS-1. An inhibitory role of skeletal muscle-derived exosomal miR-146a-5p on adipogenesis by targeting growth and differentiation factor 5 (GDF5)-PPARγ signaling has recently been reported [474]. In fact, in 3T3-L1 cells treated with *GDF5* siRNA, the expression levels of adipogenesis-related genes *GDF5*, *PPARG*, *CEBPA*, and fatty acid synthesis-related genes *CD36* (fatty acid translocase), *FABP4*, and *FASN* were significantly decreased, while in those co-treated with *GDF5* siRNA and miR-146a-5p inhibitor, the gene expressions of adipogenesis-related genes *GDF5*, *PPARG*, *CEBPA*, and fatty acid synthesis-related genes *CD36*, *FABP4*, and *FASN* were significantly increased compared with just *GDF5* siRNA treatment [470]. MiR-146a-5p also attenuates TGF-β signaling by directly targeting SMAD family member 4 (*SMAD4*) and targets TNF receptor-associated factor 6 (*TRAF6*) [475]. The addition of miR-146a-5p mimics suppressed preadipocyte differentiation, whereas a miR-146a-5p inhibitor accelerated preadipocyte differentiation [475]. TRAF6 functions as a direct E3 ligase for AKT, which is essential for AKT ubiquitination, membrane recruitment, and phosphorylation upon growth-factor stimulation and thus plays a critical role in AKT activation [476]. Preadipocytes that lack AKT exhibit differentiation defects because they fail to induce PPARγ expression at the beginning of the adipogenesis program [477,478,479]. Via inhibition of *INSR*, *GDF5*, and *TRAF6* expression, miR-146a-5p attenuates AKT/mTORC1/PPARγ signaling and preadipocyte differentiation at multiple regulatory checkpoints [472,473,474]. In contrast, Wang et al. [480] reported that miR-146a-5p via targeting ERB-B2 receptor tyrosine kinase 4 (*ERBB4*) promotes 3T3-L1 preadipocyte differentiation through the extracellular signal-regulated kinase 1/2 (ERK1/2)/PPARγ signaling pathway.

Noteworthy, miR-146a-5p was found to be highly expressed in the milk-derived EVs from sows consuming resistant starch, promoting early intestinal cell proliferation [481].

MiR-146a-5p, via targeting to silence the expression of ubiquitin ligase 3 gene ubiquitin protein ligase NEDD4-like (*NEDD4L*), inhibits Dishevelled 2 (DVL2) ubiquitination, thereby activating the Wnt pathway, promoting intestinal development [481]. In contrast, increased Wnt signaling inhibits early steps of ASC differentiation. Interestingly, exosomal miR-146a-5p from neonatal mouse cardiomyocytes promoted M1 macrophage polarization and reduced proinflammatory cytokine expression [482]. Furthermore, miR-146a-5p, which is abundant in exosomes derived from porcine primary skeletal muscle stem cells, plays a crucial role in suppressing the differentiation of adipocytes [483]. Thus, the majority of studies demonstrate anti-adipogenic effects of miR-146a-5p. Deficient transmission of HBME miR-146a-5p by FF may thus enhance early regulatory steps affecting ASC homeostasis and adipocyte differentiation.

#### 2.5.5. MicroRNA-195-5p

MiR-195-5p and miR-15b-5p are involved in the osteogenic differentiation of human adipose-derived MSCs by regulating Indian Hedgehog (IHH) expression [484]. Furthermore, miR-195-5p in human primary MSCs regulates proliferation, osteogenesis and paracrine effect on angiogenesis by targeting vascular endothelial growth factor A (*VEGFA*) [485].

Treatment of T2DM patients with pioglitazone, a potent PPARγ agonist, modified miR-195-5p in AT and circulating EVs [486]. Of note, *CCND1* is also a predicted target gene of miR-195-5p [487].

#### 2.5.6. MicroRNA-27b-3p

MiR-27b has emerged as a regulatory hub in cholesterol and lipid metabolism. In zebrafish, the depletion of miR-27b functionally promoted lipid accumulation. Sponging of miR-27 showed increased weight gain with larger fat pads, resulting from adipocyte hyperplasia. Moreover, depletion of miR-27 increased PPAR-γ, C/EBP-α, and SREBP-1c expression, contributing to lipogenesis and adipogenesis [488]. It has been shown that miR-27-3p directly targets *PPARG* [489,490] and impairs human adipocyte differentiation [489]. The anti-adipogenic effect of miR-27b-3p in human multipotent adipose-derived stem cells is due to the suppression of PPARγ [489] and C/EBPa [491]. MiR-27a-3p has also been identified as a negative regulator of adipocyte differentiation by suppressing PPARγ expression [492]. MiR-27 genes function by blocking the transcriptional induction of PPARγ and C/EBPα or by preventing preadipocytes from entering the stage of adipogenesis determination or commitment [491]. MiR-27b-3p inhibits adipogenic differentiation of human ASCs. MiR-27b-3p expression is decreased after adipogenic differentiation of human ASCs. Notably, miR-27b-3p impairs adipocyte differentiation of human AT-derived MSCs by targeting *LPL* [493]. In addition, miR-27-3p also targets the Wnt inhibitor secreted frizzled-related protein 1 (*SFRP1*) [494]. In accordance, miR-27a-3p, which shares the same seed sequence with miR-27b-3p, via targeting *SFRP1*, activates Wnt/β-catenin signaling [495,496,497].

Intriguingly, bta-miR-484, which also targets *SFRP1*, inhibits bovine adipogenesis [498]. Overexpression of bta-miR-484 in adipocytes ultimately inhibited cell proliferation and differentiation, reduced the number of EdU fluorescence-stained cells, increased the number of G_1_ phase cells, reduced the number of G_2_ and S phase cells, and downregulated the expression of proliferation markers (CDK2 and PCNA) and differentiation markers (C/EBPα, FABP4, and LPL) [498]. *SFRP1* regulates AT expansion and is dysregulated in severe obesity [499]. Constitutive ectopic expression of SFRP1 is pro-adipogenic and inhibits Wnt/β-catenin signaling in 3T3-L1 adipocytes [500]. Thus, miR-27b-3p not only directly targets *PPARG* [492] but, via suppression of *SFRP1*, increases Wnt signaling and synergistically enhances the inhibitory effect of Wnt signaling on *PPARG*. Of note, effective osteogenic priming of MSCs through locked nucleic acid-antisense oligonucleotides (LNA-ASOs)-mediated *SFRP1* gene silencing has been reported [501].

#### 2.5.7. MicroRNA-34a-5p

MiR-34a-5p inhibits the differentiation of human ASCs [502]. Overexpression of miR-34a-5p decreases cell proliferation and the expression of various cell cycle regulators such as CDK2, CDK4, CDK6, cyclin E, and cyclin D, as well as surface expression of stem cell markers, including CD44 [502,503]. It has been shown that miR-34a-5p targets *PDGFRB*. MiR-34a-5p can inhibit PDGFRβ protein expression at a post-transcriptional level, suppress Ras/MAPK signaling pathways, and downregulate expression of cell cycle proteins at the G_0_/G_1_ phase, such as cyclin D1, CDK4, and CDK6 [504]. PPARγ transcriptionally activates PDGFRβ and VEGF. PPARγ regulates APC niche occupancy and WAT vascular expansion. VEGF is a transcriptional target of PPARγ in APCs. APC expression and activation of PDGFRβ promote the recruitment and retention of APCs to the niche, whereas inhibition of PDGFRβ disrupts APC niche contact, thus blocking AT expansion [343]. Benvie et al. [505] recently showed that activating PDGFRβ in juvenile mice blocks beige fat formation. Notably, *FTO* dependent m^6^A RNA-mediated demethylation activates PDGFR-β/ERK signaling [435].

MiR-34a-5p also targets C1q/tumor necrosis factor-related protein-9 (CTRP9). Down-regulation of miR-34a-5p and upregulation of CTRP9 promote ASC proliferation and migration [506]. In addition, *FLOT2*, a protein enriched in pro-adipogenic ECM orchestrating ECM to preadipocyte signaling [242], is a predicted target of miR-34a-5p [243]. Recruitment of *EZH2* and upregulation of H3K27me3 at the *MIR34A* promoter region inhibits miR-34a expression [507] and represses *WNT1*, *WNT6*, *WNT10A*, and *WNT10B* genes in preadipocytes and during adipogenesis [204]. As already outlined, increased FF-induced S6K1 activity enhances nuclear recruitment of *EZH2*, thereby potentially attenuating miR-34a-5p expression [258]. Furthermore, overstimulated insulin/IGF-1/PI3K/AKT signaling reduces cellular levels of p53 [400]. Importantly, *MIR34* expression is activated by p53 [508,509]. It is thus conceivable that FF with increased AKT/mTORC1 activity and reduced p53 signaling attenuates the expression of anti-adipogenic miR-34a-5p. In addition, miR-34a-5p-deficient formula may aggravate ASC proliferation and WAT adipogenesis.

#### 2.5.8. MicroRNA-612

MiR-612 may also be involved the regulation of adipogenesis. *TP53* is a putative target gene of miR-612 [510]. MiR-612 has been discussed as a prospective biomarker of response to specific weight-loss diets [510]. Of note, miR-612 overexpression downregulates *VEGFA* [511] and *AKT2* [512,513].

#### 2.5.9. MicroRNA-148a-3p

MiR-148a-3p is highly expressed in colostrum exosomes, especially under conditions of preterm birth, and declines in mature milk [169,170]. MiR-148a-3p, via targeting the WNT signaling inhibitor Dickkopf 1 (*DKK1*), promotes osteogenesis [514]. *DKK1* interacts with LRP5/6-receptors inhibiting Wnt signaling. *DKK1*, produced from MSCs, inhibits osteogenesis but induces adipogenesis, effectively switching the MSC differentiation pathway by inhibiting the Wnt/β-catenin pathway [515]. Fan et al. [516] demonstrated that increasing the expression of *DKK1*, which competitively binds with LRP5 to inhibit the Wnt/β-catenin pathway, reduced the inhibition of adipogenesis by Wnt signaling. *DKK1* is abundantly expressed in the early stages of adipogenesis and decreases during the late stages [517]. Lu et al. [518] demonstrated that recombinant human *DKK1* promotes ASC differentiation. In contrast, Tian et al. [519] reported that supplementation of miR-148a-3p blunted osteoblast differentiation via targeting lysine-specific demethylase 6b (*KDM6B*), a recently identified regulator of osteoblast differentiation. Nevertheless, the majority of studies imply that miR-148a-3p promotes osteoblast but inhibits ASC differentiation. Of note, the MAX protein (also known as MYC-associated protein X), the most conserved dimerization component of MYC, is critically involved in the stimulation of MCE, and has been predicted to be a conserved target gene of miR-148a-3p and miR-22-3p [520].

#### 2.5.10. MicroRNA-155-5p

MiR-155-5p has been detected in higher concentration in human colostrum and colostrum exosomes compared to mature human milk and belongs to the group of early immune-regulating miRs [521,522,523,524]. MiR-155-5p has been demonstrated to directly target *FTO* [525,526] and *CEBPB* [527], two critical regulators of ASCs and early adipogenesis.

#### 2.5.11. MicroRNA-30-5p

Wu et al. [447] reported higher expression levels of miR-30b-5p in colostrum compared to mature milk. Yun et al. [453] observed that exosomes of human colostrum express higher levels of miR-30a-5p compared to colostrum, in accordance with Verma et al. [454]. Remarkably, miR-30a-5p and miR-30b-5p share the same conserved seed sequence with human *FTO* [147,168]. In zebrafish, miR-30b-5p suppressed *FTO* and lipogenesis [147]. The miR-30 family plays an important role in controlling proliferation and differentiation of intestinal epithelial cell by targeting the transcription factor sex-determining region Y (SRY)-box 9 protein (SOX9) [528]. Le Guillou et al. [529] showed that feeding mice pups with modified milk resulting from a single miR-30b deregulation in the mammary gland can lead to long-term changes in the offspring, with consequences to growth and intestinal physiology. Stöckl et al. [530] provided evidence in rat MSCs that the major transcription factor for adipogenic differentiation, C/EBPβ, is repressed after silencing SOX9. A delicate balance of SOX9 levels is apparently involved in proper adipogenic, chondrogenic, and osteogenic progenitor cell differentiation [530]. In fact, inhibition of SOX9 specifically represses C/EBPβ protein synthesis as well as PPARγ, SREBF1, and FASN [530]. Notably, a low SOX9 dose delays the progression of adipogenic differentiation of MSCs [530]. Conversely, SOX9 downregulation has been reported to be required for adipocyte differentiation [531,532,533]. Preadipocyte factor-1 (PREF-1), a transmembrane epidermal growth factor-like domain-containing protein encoded on the *DLK1* gene, is highly expressed in 3T3-L1 preadipocytes, but is undetectable in mature adipocytes. It inhibits adipocyte differentiation through upregulating SOX9 expression [531,532,533]. SOX9 directly binds to the promoter regions of *CEBPB* and *CEBPD* to suppress their promoter activity preventing adipocyte differentiation [531,532,533]. *DLK1* expression is also regulated by miR-15a [534], pointing to a complex regulatory network fine-tuning PREF-1/SOX9 expression in MSCs. Remarkably, 3T3-L1 cells secrete higher concentrations of exosomes enriched in PREF-1 and PPARγ prior to adipogenesis [535].

Taken together, the great majority of highly expressed early miRs in colostrum and those miRs reported to decline during the course of lactation towards mature milk exert inhibitory effects on ASC development and adipogenesis of WAT but promote BAT and osteogenesis (Table 2). During the early stages of lactation, a complex interacting anti-adipogenic miR network apparently operates to limit ASC and WAT development but promotes BAT and thermogenesis, which is of higher biological importance for the neonate, especially the preterm infant sensitive to cold. The absence of anti-adipogenic miRs in formula apparently is a severe regulatory fault promoting early excessive ASC proliferation and WAT adipocyte cellularity but impairing BAT development, an early risk constellation promoting obesity. Indeed, breastfed neonates delivered at term lose 6.6% of median birthweight during their first week of life compared to formula-fed term neonates (median weight loss 3.5%) [536].

Milk-derived exosomes may converge with adipocyte-derived exosomes and their miR cargos in the regulation of adipogenesis [537,538]. Shi et al. [539] monitored miR levels in human adipose-derived MSCs (hMSCs-Ad), human stromal vascular cells (SVCs), and differentiated adipocytes, and detected 42 differently expressed miRs (meta-signature miRs) in mature adipocytes compared to SVCs or hMSCs-Ad. Meta-signature miRs specific for adipogenesis included let-7 family, miR-15a-5p, miR-27a-3p, miR-106b-5p, miR-148a-3p, and miR-26b-5p, respectively [539]. MiR-148a-3p, by targeting *WNT1*, an inhibitor of adipogenesis, promotes hMSCs-Ad differentiation [540]. Röszer [541] defined the miR cargo of EVs secreted by mouse adipocytes on postnatal day 6, when adipocytes are lipolytic and thermogenic, and on postnatal day 56, when adipocytes have active lipogenesis. The most abundant miR of mouse adipocyte-derived EVs was miR-148a-3p, which increased during murine adipocyte maturation.

Monocyte chemoattractant protein-induced protein 1 (MCPIP1) impairs adipogenesis in 3T3-L1 cells [542] and promotes M2 macrophage polarization [543]. Knockdown of MCPIP1 results in an upregulation of C/EBPβ and PPARγ mRNAs [542]. MCPIP1 possesses the N-terminus of the PilT protein that has RNase properties and degrades transcripts coding for inflammation- and differentiation-related proteins, and suppresses miR biogenesis [544,545]. MCPIP1 overexpression results in modulated levels of 58 miRs in adipocytes on day 2 of differentiation [544]. Figure 5 represents the expected differences of miR-regulated gene expression between FF (Figure 5A) and BF (Figure 5B).

Taken together, miRs play a key role in the regulation of ASC homeostasis and the promotion or suppression of adipogenesis. Adequate levels of exosomal milk miRs provided by natural BF provide a complex miR network for the appropriate postnatal adjustment of ASC numbers and adipocyte differentiation during the early stage of lactation. In contrast, the postnatal absence of anti-adipogenic exosomal milk-derived miRs by artificial FF may disturb the proper development of the postnatal adipocyte niche, leading to an increase in the total number of ASCs and enhancing adipocyte cellularity.

### 2.6. Postnatal Adipose Progenitor Cell Development in the Vascular Niche and Obesity Risk

The complex process of adipogenesis is not only restricted to the development of adipocytes, the predominant cell type of WAT, but critically involves the stromal vascular fraction (SVF) with stromal, mural (endothelial cells, pericytes, smooth muscle cells, fibroblasts), and immune cells (macrophages, T cells) that all control ASC commitment, adipocyte differentiation, and WAT development and homeostasis [278,462,463]. The primary development of AT and adipogenesis occurs during the critical phases of pregnancy and lactation, during which alterations in the perinatal environment likely imprint lasting changes in the characteristics of offspring AT [546,547]. Both in rodents and humans, before and after birth, angiogenesis appears to be closely coordinated in time and space with the formation of fat cell clusters [548]. Pioneering electron microscope studies already showed vascular-residing cells peeling away from the blood vessel as they transitioned into lipid-filled cells or adipocytes [549]. Recent genetic studies have positioned APCs to the vascular niche [550,551]. These APCs reside in perivascular positions along blood vessels within the adipose tissue. Intriguingly, Jiang et al. [551] identified two progenitor populations that give rise to adipocytes designated as developmental progenitors for adipose organogenesis and adult progenitors for adipose homeostasis. Both progenitor compartments express PPARγ and produce adipocytes, yet have distinct functional and molecular properties and even reside in distinct anatomical niche localities. Remarkably, adult progenitors, which derive from a perivascular position and play key roles in fat depot formation and AT homeostasis, appear even earlier than developmental progenitors [551]. The appropriate interaction between the cellular and matrix components along with proper angiogenesis are mandatory for the development of AT [552]. For instance, postnatal epididymal adipose tissue (EAT) in mice is generated during the first 14 postnatal days. From postnatal day 1 (P1) to P4, EAT is composed of multipotent progenitor cells that lack adipogenic differentiation capacity in vitro [553]. The SVF of AT is the preferred area of residing ASCs [100,101,102,103,104,551,552,553,554].

Upon adipogenic signals, these cells peel away from the SVF, becoming lipid-filled adipocytes loosely associated with the vascular unit [105,553,555]. Most adipocytes descend from a pool of these proliferating progenitors that are already committed, either prenatally or early in postnatal life. These progenitors reside in the mural cell compartment of the adipose vasculature [105]. Notably, the blood vessel niche is the microenvironment controlling APC number and AT mass [105,555]. In fact, Berry et al. [105] described the intimate interaction of vascular endothelial cell and closely attached ASCs as “blood brothers”. Blood vessels are the critical niche stimulating APC expansion and differentiation [105,555] (Figure 1). Uhrbom et al. [556] recently provided evidence that ASCs are sexually dimorphic cells that serve a dual role as adipocyte precursors and fibroblast-like cells that shape the AT’s extracellular matrix in an organotypic manner. The authors conclude that ASCs are distinct from mural cells, and that the state of commitment to adipogenic differentiation is linked to their anatomic position in the microvascular niche [556].

#### 2.6.1. ETS Proto-Oncogene 2

Several studies have shown that AAs are key regulators in maintaining vascular homeostasis by modulating endothelial cell (EC) proliferation, migration, survival, and function [557]. The transcription factor ETS proto-oncogene 2 (*ETS2*) is induced early during adipogenesis in vitro, and its expression is enriched in the SVF of WAT in vivo, consistent with its expression in early adipogenesis in vitro [558]. Enrichment of *ETS2* in adipocyte progenitor cells in vivo suggests that *ETS2* plays a functional role in adipocyte differentiation, whereas *ETS2* knockdown in 3T3-L1 cells has been shown to impair adipogenesis through a reduction of clonal expansion [558]. Of note, *ETS2* expression is modified by m^6^A demethylation [559]. Overstimulated *FTO* expression in APCs by enhanced circulatory transfer of BCAAs may promote m^6^A-mediated *ETS2* expression. In contrast, *ETS2* mRNA is a predicted target of miR-22-3p [560], the most abundant miR of milk exosomes provided during early stages of lactation [453,454] and preterm delivery [169]. Milk exosomal miRNA-22-3p may not only adjust the appropriate calibration of *FTO* [168,230], but also of *FTO*’s potential downstream target *ETS2* [560].

#### 2.6.2. Platelet-Derived Growth Factor Receptors

APCs express platelet-derived growth factor receptors (PDGFRs), PDGFRα and PDGFRβ [561,562]. The PDGFRα/PDGFRβ signaling balance determines progenitor commitment to beige (PDGFRα) or white (PDGFRβ) adipogenesis [561]. Sun et al. [562], using mosaic lineage labeling, showed that adipocytes are derived from the lineage during postnatal growth and adulthood, whereas adipocytes are only derived from the mosaic PDGFRB lineage during postnatal growth. Downregulation of PDGF signaling is regarded to be a critical event in the transition from APCs to adipocytes. PDGFRβ is a marker of white adipocyte progenitors involved only in postnatal adipocyte development [561,562]. PDGFRβ^+^ mural preadipocytes contribute to adipocyte hyperplasia induced by HFD in adult mice [551]. Recent evidence indicates that *FTO* via m^6^A demethylation of *PDGFRB* mRNA enhances PDGFRβ expression in acute myeloid leukemia cells [563]. In contrast, *FTO* deletion enhanced the expression of *PDGFRA* mRNA and PDGFRα protein in murine neural stem cells [564]. Increased *FTO* activity may thus alter the ratio of PDGFRα/PDGFRβ. *FTO* appears to contribute to the expression of PDGFRβ on adipocyte progenitors of WAT. In contrast, the Notch–PDGFRβ axis suppresses brown APC differentiation in early postnatal mice [565]. Further information regarding adipose precursor populations that contribute to the physiological postnatal recruitment of white, brown, and beige adipocytes and novel insights into RNA m^6^A-mediated post-transcriptional regulation of adipogenesis has recently been reviewed extensively [566,567].

#### 2.6.3. α-Smooth Muscle Actin

ASCs originate from perivascular cells and congregate around blood vessels (Figure 1) [568]. Notably, α-smooth muscle actin (α-SMA)-GFP-positive cells congregate around the blood vessels and have multilineage differentiation ability [568]. It has been shown that *FTO* promotes fibroblast migration as well as the expression of α-SMA [569]. In contrast, in isolated smooth muscle cells of *FTO*^−/−^ mice, a significant decrease in *ACTA2* was observed, further confirmed with *FTO* siRNA [570]. Notably, *ACTA2* is a target of miR-27a-3p and miR-27b-3p [571,572,573]. MiR-27b-3p is one of the critical miRs expressed in early lactation-derived milk exosomes suppressing *PPARG* and *SFRP1*, affecting ASC determination and adipogenesis [489,490,494].

A deeper understanding of adipocyte precursor lineages and their different regulatory mechanisms during various ages, including the postnatal period [574], and the role of micro-environmental cues that influence cell identity and cell behavior at various junctures in adipocyte lineage development [575] require further investigation. However, *FTO* overexpression induced by uncontrolled high protein intake by FF combined with the deficiency of anti-adipogenic early lactation-derived exosomal miRs (Table 2) may disturb the regulatory homeostasis of cells of the SVF, affecting ASC determination, differentiation, and adipogenesis originating in the vascular niche.

#### 2.6.4. Vascular Endothelial Growth Factor

Within the adipose tissue stem cell niche, diverse cell types, such as endothelial cells, immune cells, mural cells, and adipocytes, intricately regulate the function of adipocyte precursors [576]. Yet, the nutritional signals that regulate adipose vascular niche formation and APC niche interaction are less studied. Jiang et al. [343] showed that the assembly and maintenance of the adipose vascular niche are controlled by a signaling network involving PPARγ, PDGFRβ, and VEGF. Experimental evidence supports the view that APCs direct adipose tissue niche expansion via a PPARγ-initiated PDGFRβ and VEGF transcriptional axis. PPARγ regulates APC niche occupancy, WAT vascular expansion, and WAT APC niche interaction and expansion. PDGFRβ, a direct transcriptional target of PPARγ, regulates niche formation and maintenance, regulates APC niche retention, and restores niche function under PPARγ deficiency. APC VEGF mediates PPARγ-induced niche expansion, stimulating WAT niche expansion [308]. Not only the expression of *PPARG* and *PDGFRB*, but also of *VEGFA* [577,578,579], is epigenetically regulated by the m^6^A methyltranscriptome. In primary mouse bone marrow-derived macrophages, knockdown of *FTO* expression by specific siRNA significantly dampened macrophage-mediated VEGFA release [577]. *FTO* supports cancer-associated fibroblast-mediated angiogenesis through the activation of early growth response 1 (EGR1) and VEGFA [578]. Increased *FTO* induces ocular angiogenesis by controlling EC function in an m^6^A-YTHDF2-dependent manner [579].

It is conceivable that increased protein intake by FF with *FTO* overexpression upregulates VEGF, promoting PPARγ-induced WAT niche expansion [308], a critical step exacerbating ASC nursing conditions and development. For instance, HFD feeding in mice rapidly and transiently induces proliferation of APCs within WAT to produce new adipocytes, specifically in the perigonadal visceral depot in male mice, consistent with the patterns of obesogenic WAT growth observed in humans [580]. Conversely, reduced protein intake and aging affects the sustainment of hematopoiesis by impairing bone marrow MSCs in mice [581].

#### 2.6.5. FTO- and LINE1 m^6^A RNA-Mediated Chromatin Opening and Gene Expression

Long interspersed nuclear element-1 (LINE1) is a retrotransposon group that constitutes 17% of the human genome and shows variable expression across cell types [582]. Cells use intragenic LINE1s as cis-regulatory elements within gene bodies to modulate gene expression [583]. Recent evidence indicates that LINE1 transcription activates long-range gene expression [582]. Of note, LINE1s can physically contact their distal target genes, with these interactions becoming stronger upon LINE1 activation and weaker when LINE1 is silenced [582]. Recent studies have demonstrated an important role for m^6^A in regulating mouse ESC (mESC) fate and controlling early mammalian embryonic development [584]. Wei et al. [585] recently demonstrated that *FTO* mediates m^6^A demethylation of LINE1 RNA in mESCs regulating LINE1 RNA abundance and the local chromatin state, which in turn modulates the transcription of LINE1-containing genes. They observed LINE1 RNA–chromatin interaction, as well as co-localization and binding of LINE1 RNA and *FTO* in mESCs, supporting LINE1 RNA as a physiological target of *FTO* [585]. Activation of LINE-1 regulates global chromatin accessibility at the beginning of development and indicates that retrotransposon activation is integral to the developmental program [586,587]. In 2011, Prokesch et al. [588] were the first to report that retrotransposed genes, termed adipocyte-related X-chromosome-expressed sequence 1 (ARXES1) and ARXES2, are involved in the regulation of adipogenesis. In particular, the two paralog genes, which arose by retrotransposition of the parental gene signal peptidase complex, subunit 3 (SPCS3), followed by a segmental duplication event, are upregulated during adipogenesis in different model systems by C/EBPα and PPARγ/RXRα through proximal promoter sites and, possibly, a distant enhancer region. Knockdown of ARXES1 and ARXES2 abolished differentiation of 3T3-L1 preadipocytes, while knockdown of SPCS3 had no effect on adipogenesis. ARXES mRNAs are highly expressed in AT and strongly upregulated during adipogenesis [588]. Silencing of ARXES expression in MSCs attenuated adipogenesis while augmenting differentiation to osteoblasts [587]. Recently, Chen et al. [589] observed that zinc finger protein 30 (ZFP30) promotes adipogenesis by directly targeting and activating a retrotransposon-derived *PPARG2* enhancer. ZFP30 recruits the co-regulator KRAB-associated protein 1 (KAP1), which acts as a ZFP30 co-activator in adipogenesis [589]. The PPARγ isoform PPARγ2 is a powerful modulator of MSC-related gene expression [590] and is expressed in cells of the adipocyte lineage, serving as an essential regulator of early and terminal adipocyte differentiation [591,592,593,594].

In conclusion, FF-induced upregulation of *FTO* via enhanced LINE1 expression may excessively open chromatin and activate adipogenic transcription (*PPAR*γ*2*), which may overstimulate postnatal ASC proliferation and differentiation.

### 2.7. FTO-Dependent Immune Cell-Mediated ASC Differentiation

Compared to exclusive BF, significant overexpression of *FTO* in PBMCs of formula-fed infants compared to BF has been observed [110]. PBMCs are primarily composed of lymphocytes (T cells, B cells, and NK cells), monocytes, and dendritic cells. In humans, lymphocytes typically make up 70–90%, monocytes 10–20%, and dendritic cells only 1–2% [595]. Immune cells are the most diverse cell populations in AT and play essential roles in regulating AT function through interactions with adipocytes as well as adipocyte progenitors. Macrophages, mast cells, innate lymphoid cells, and T cells are all involved in the regulation of APC proliferation, differentiation, and lineage commitment [596,597] (Figure 1).

#### 2.7.1. CD40 Antigen and CD40 Antigen Ligand

While the majority of studies have focused on the influence of MSCs and ASCs on T cells [598,599,600,601], the impact of circulating T cells on ASCs in the vascular niche has been less characterized. Intriguingly, Gregersen et al. [602] demonstrated that mice displaying an altered composition of circulating T cells increased T cell activation in VAT, pointing to a crosstalk between activated T cells and adipogenesis. There is recent interest in the role of m^6^A modifications in the regulation of immune cells [603]. In fact, van Vroonhoven et al. [604] showed that m^6^A levels modified by *FTO* directly regulate CD40L expression in CD4^+^ T lymphocytes. YTHDF2 binding to m^6^A specific sequences on the *CD40L* mRNA apparently promotes its degradation [604]. CD40 is expressed on APCs and adipocytes. *CD40* mRNA and protein expression increase during adipocyte differentiation and correlate with BMI [605]. Furthermore, CD40L-expressing CD4^+^ T cells prime adipose-derived stromal cells to produce inflammatory chemokines [606].

The interaction between CD40 and CD40 ligand (CD40L), a crucial co-stimulatory signal for activating adaptive immune cells, plays a critical role in atherosclerosis [607]. Adipocytes control hematopoiesis and inflammation through CD40 signaling [608]. Importantly, Missiou et al. [609] demonstrated that stimulation with CD40L resulted in enhanced activation of C/EBPα and PPARγ and promoted adipogenesis of preadipose cells in the presence and absence of standard adipogenic conditions. CD40L^−/−^ mice are protected from weight gain [610,611], and in BAT exhibit increased mRNA levels of UCP-1 indicating that a deficiency of CD40L promotes thermogenesis [610].

CD40L is also expressed on endothelial cells, monocytes, and macrophages [612], which may interact with ASCs as well. Conditioned media from CD40L-pretreated adipocytes provoked elevated migration of mononuclear cells and increased the expression of inflammatory genes in bone marrow-derived mononuclear phagocytes (BMDM), shifting them to an M1-like pro-inflammatory phenotype [613].

MiR-3168 has been identified as the most upregulated miR in preterm HBMEs and has been implicated in playing a key role in neural stem cell differentiation [614]. Notably, miR-3168 targets *CD40LG*, along with other miRs expressed during early lactation (miR-623, miR-146a-5p, miR-149-3p) [615]. Thus, FTO-mediated upregulation of CD40L [604] enhancing the activation of C/EBPα and PPARγ in preadipocytes [609] may be counterbalanced by miRs expressed during preterm and early stages of lactation.

Taken together, FF-induced upregulation of *FTO* may enhance T cell and monocyte/macrophage CD40L expression promoting the differentiation of CD40-expressing APCs promoting adipogenesis, whereas suppression of CD40L via early lactation-derived exosomal miRs may promote BAT and thermogenesis (Figure 6).

#### 2.7.2. Vascular Cell Adhesion Molecule 1 and Intercellular Adhesion Molecule 1

Recent evidence indicates that *FTO* is also involved in regulating endothelial function [616]. Knockdown of *FTO* attenuated the expression of vascular cell adhesion molecule 1 (VCAM-1) and intercellular adhesion molecule 1 (ICAM-1), as well as the adhesion of monocytes to endothelial cells [616]. VCAM-1 is responsible for the adhesion of various immune cells to the vascular endothelium, while ICAM-1 signaling recruits inflammatory immune cells like macrophages and granulocytes, aiding their movement into tissues and the adipose vascular niche [617,618]. Therefore, *FTO* plays a role in endothelial cell function [616] and VEGFA-mediated angiogenesis [577,578], potentially enhancing the transmigration of immune cells into the adipose vascular niche and impacting ASC development. Table 3 summarizes *FTO* in m^6^A-mediated modifications of key molecules involved in ASC determination, adipocyte stem cell niche formation, and adipogenesis.

#### 2.7.3. *FTO* and Macrophage–Adipocyte Stem Cell Interaction

Remarkably, macrophages coincidentally appear during WAT development. Recent evidence indicates that macrophages can influence stem/progenitor cell properties, functions, and destiny, and vice versa [619,620,621,622]. Co-culture of adipocytes with AT macrophages (ATMs) and ASCs increased the formation of new preadipocytes, thereby increasing lipid accumulation and C/EBPα and PPARγ gene expression, suggesting that preadipocytes may originate in part from ATMs [623]. Nawaz et al. [624] recently reported that the depletion of CD206^+^ M2-like macrophages resulted in the enhanced generation of smaller adipocytes, improving insulin sensitivity and proliferation of APCs. M2-like macrophages in AT regulate systemic glucose homeostasis by inhibiting adipocyte progenitor proliferation via the CD206/TGFβ signaling pathway [625].

RNA m^6^A modification significantly affects macrophage function from the perspective of their development, activation, polarization, pyroptosis, lipid uptake, and cholesterol efflux [626]. It has recently been observed that RNA m^6^A and 5hmC influence gene expression programs during macrophage differentiation and polarization [627]. Gu et al. [628] found that *FTO* silencing significantly suppressed both M1 and M2 polarization, inhibited the NF-κB signaling pathway, and reduced the mRNA stability of STAT1 and PPARγ via YTHDF2 involvement, thereby impeding macrophage activation. Hu et al. [629] showed that miR-495-mediated silencing of *FTO* induced the transformation of macrophages into M1-type pro-inflammatory macrophages. As shown by CRISPR/Cas9 system knockout of the *FTO* gene in macrophages, *FTO* affects the stiffness-controlled macrophage inflammatory response by sustaining the negative feedback generated by suppressor of cytokine signaling 1 (SOCS1) [630]. METTL14, YTHDF1, and *FTO* regulate *SOCS1* m^6^A methylation to sustain an appropriate SOCS1 level so that the negative feedback loop in lipopolysaccharide (LPS)/Toll-like receptor 4 (TLR4) signaling is maintained to control macrophage inflammatory response [631]. Diabetes induces a shift in macrophage polarization towards a pro-inflammatory M1 phenotype, which is associated with a reduction in m^6^A modification levels. Feng et al. [632] also found a relationship between *FTO* deficiency and M1 macrophage polarization. Monocytes and macrophages constitutively express CD40 and are capable of a robust response to CD40 ligation, resulting in the induction or enhancement of expression of pro-inflammatory genes [633] (Figure 6).

Taken together, *FTO* affects endothelial cell adherence of monocytes and macrophage polarization. However, the potential biological effects of FF on FTO-related macrophage–ASC interactions in the SVF during the postnatal period require further investigation.

### 2.8. Postnatal Increase in Adipocyte Cellularity Enhances Obesity Risk

In mice, adipocytes in the gonadal WAT differentiate postnatally between birth and sexual maturation, whereas all adipocytes in the subcutaneous AT start to differentiate between E14 and E18, but the differentiation takes much longer and finishes postnatally [634]. In humans, adipogenesis and development of AT predominantly occur before birth. The onset of adipogenesis typically begins around the 14th to 17th week of gestation, initially forming clusters of fat lobules [635]. Following birth, both the number and size of adipocytes increase. In a cross-sectional study involving children, adipocyte size reaches adult levels between 6 months and 1 year of age and then gradually decreases between one and two years [636]. Deviations from this normal development were observed in obese children shortly after 1 year of age. By 11 years of age, obese children exceeded the mean cell number found in nonobese adults. Indeed, obese subjects displayed more rapid and earlier elevations in both cell number and size, which were maintained throughout the study [636]. The data indicate that the rate and type of AT cellular development that one encounters in children may play a role in the development of the enlarged fat depots found in obese subjects [636]. However, the number of fat cells stays constant in adult lean and obese individuals, even after marked weight loss, indicating that the number of adipocytes is set during childhood and adolescence [637]. Thus, the quantity of adipocytes within a specific depot is predominantly established during early life and remains relatively constant throughout adulthood [637,638,639,640,641,642,643,644,645]. These data support the view that the postnatal period is a critical and vulnerable phase determining the fate of ASCs and the lifetime risk of obesity.

## 3. Discussion

Our review presents an extensive epigenetic framework explaining the “early protein hypothesis” of obesity [9] primarily based on increased mTORC1/S6K1 activation [60,90], which has recently been appreciated in the field of pediatric research [41]. As demonstrated, we are able to link overactivated mTORC1/S6K1 signaling to downstream transcriptional and epigenetic changes that accelerate ASC expansion in adipose vascular niches during postnatal WAT development. Postnatal protein excess enhances upstream activators of mTORC1/S6K1, including insulin [61,78,79,80], IGF-1 [61], BCAAs [61,80], and the amino acid sensor *FTO* [110,117], respectively. mTORC1/S6K1 activate the protein expression of the histone modifier *EZH2* [646]. Activated S6K1 enters the nucleus and recruits *EZH2* to H3K27, resulting in H3K27me3-mediated *WNT* gene suppression [258,259]. Importantly, *WNT* suppression is the critical epigenetic switch enhancing MSC fate determination towards adipogenesis [185,204,252]. Thus, excessive postnatal protein intake elicits a central signaling hierarchy of mTORC1/S6K1/*EZH2*-mediated *WNT* gene suppression, promoting ASC commitment and increasing ASC numbers (Figure 2). In fact, activation of *EZH2* plays a key role in directing MSCs to adipocyte lineage commitment and suppression of osteogenesis, and vice versa [647]. *EZH2* directly increased H3K27me3 levels on promoters of *WNT1*, *WNT6*, and *WNT10A* to silence *WNT* gene transcription, shifting MSC cell lineage commitment to adipocyte [204,647]. In contrast, suppression of *EZH2* has been shown to prevent the shift of MSC fate to adipocyte, enhancing bone formation [648,649]. Furthermore, the *EZH2* inhibitor GSK126 inhibits the differentiation of MEFs into white adipocytes but promotes their differentiation into brown/beige adipocytes [256] and alleviates the obesity phenotype by promoting the differentiation of thermogenic beige adipocytes in diet-induced obese mice [648].

Activated *FTO* stimulates amino acid-mediated mTORC1/S6K1-*EZH2* signaling [111,112], but through its direct impact on the m^6^A-RNA methylome, it epigenetically augments the expression of key transcription factors including PPARγ and SREBF1, paving the way for key adipogenic effectors of ASCs [130] and adjacent cells of the ASC microenvironment in the SVF (Table 3). The donation of physiological quantities of protein by exclusive BF acts antagonistically to the *FTO* gain-of-function polymorphism (rs9939609 risk allele) [153,154].

Among mTORC1-S6K1-*EZH2*-mediated changes in histone modifications and *FTO*/m^6^A-mediated changes in mRNA expression, HBM-derived miRs, which are deficient in artificial formula, represent a third epigenetic regulatory layer tuning the early mRNA expression of *FTO*, key adipogenic transcription factors, and upstream regulators of canonical WNT signaling. We suggest that HBM-derived miRs of the early lactation period attenuate WAT but promote BAT development, thereby maintaining appropriate thermogenesis.

In accordance with recent literature evidence, we would like to discuss important observations that fit well into the presented epigenetic network of upregulated *FTO*/mTORC1/S6K1/*EZH2* and downregulated Wnt signaling due to excessive protein intake by FF as the major pathway promoting ASC commitment and expansion.

Excessive intake of cow milk (milk protein) after the weaning period enhanced the proliferation of subcutaneous porcine ASCs [210] and increased murine pS6K1 expression in WAT [211]. Emerging genetic evidence in both humans and mice suggest central roles for Wnt signaling in body fat distribution, obesity, and metabolic dysfunction [650]. In 2005, Arango et al. [651] provided the first molecular evidence linking suppressed canonical Wnt signaling by disruption of β-catenin expression in MSCs to a switch from myogenesis to adipogenesis in mice. The suppression of Wnt signaling has a profound impact on MSC fate determination promoting adipogenesis but suppressing osteogenesis [273,276,652,653]. During lineage commitment of MSCs into preadipocytes (ASCs), Wnt signaling is inactivated [185,650,654]. Active Wnt signaling maintains a pool of non-differentiated progenitor cells [655], whereas suppressed Wnt signaling promotes adipocytic lineage commitment [185,647,650,654,655]. Notably, Wnt signaling actively inhibits the expression of PPARγ and C/EBPa in preadipocytes [185,186,252] (Figure 3), underlining the important involvement of Wnt signals in controlling the function of key adipogenic transcription factors.

In growing–finishing pigs, dietary L-arginine supplementation increases intramuscular fat content [656]. Arginine is known to activate mTORC1 [657]. Remarkably, PPARγ and C/EBPα were upregulated by arginine supplementation, whereas activation of the Wnt/β-catenin signal pathway by using lithium chloride significantly attenuated arginine-induced upregulation of PPARγ and increased the phospho-β-catenin levels [656]. Importantly, Wnt signaling suppresses the expression of *FTO* [264], whereas PPARγ and C/EBPα both promote the expression of *FTO* [173,187]. In a feed-forward loop, *FTO* via m^6^A RNA demethylation enhances further expression of PPARγ [129,173,174] and C/EBPα [129] (Figure 3).

FF with increased S6K1-mediated *EZH2*-induced H3K27me3-mediated *WNT* suppression might thus overstimulate MSCs to enter ASC commitment, paving the way to obesity, whereas BF apparently controls the appropriate postnatal magnitude of Wnt signaling in the ASC vascular niches. Notably, early epigenetic silencing of Wnt signaling via increased expression of the Wnt inhibitor secreted frizzled-related protein 5 (SFRP5) enhances AT expansion and increases the susceptibility to HFD for weight gain even before mice are fed an HFD [658]. In addition, excessive protein intake increases the expression of *FTO* [108], which is regulated by the availability of essential amino acids [109]. Cheshmeh et al. [110] demonstrated in PBMC of 5–6-month-old FF human infants that *FTO* is excessively overexpressed compared to BF. FF-mediated upregulation of *FTO* disturbs the epigenetic calibration of multiple adipogenic genes (Table 3), whose expression is controlled at the level of m^6^A RNA methylation. Translational evidence collected in our review demonstrates that early postnatal protein intake has a regulatory impact on ASC homeostasis in the SVF.

Furthermore, ASC gene expression of adipogenic transcription factors, the regulation of MCE, and further differentiation of preadipocytes into mature adipocytes are controlled by a network of miRs. Presented evidence in the SVF points to the postnatal contribution of colostrum and milk exosomal miRs to the proper adjustment of ASC proliferation and differentiation. Remarkably, predominant miRs detected during the early lactation period, as well as under preterm birth conditions, suppress genes involved in WAT development but enhance BAT formation and thermogenesis. Thus, HBME miRs may fine-tune physiological requirements for the postnatal balance of WAT/BAT development. In fact, suppression of *FTO* upregulates UCP-1, the development of BAT and thermogenesis [165,166,167], important physiological demands for energy and temperature regulation of the newborn and especially the preterm infant.

Fibroblast growth factor 21 (*FGF21*) is another key mediator of energy metabolism, inducing browning of AT [659,660,661]. Prolonged BF protects from obesity by the hypothalamic action of hepatic *FGF21* [662]. Notably, mice fed a low-protein diet exhibited *FGF21*-dependent browning in subcutaneous WAT with increased expression of UCP-1 [663]. Wali et al. [664] recently confirmed that FF results in significantly higher serum concentrations of IGF-1, insulin, and C-peptide concentrations compared to BF. In contrast, *FGF21* levels were generally higher in breastfed infants, pointing to *FGF21* as a possible novel mediator underpinning the early protein hypothesis [664]. According to a genome-wide meta-analysis of a population-based discovery cohort, rs838133 in *FGF21*, rs197273 near TRAF family member-associated NF-kappa-B activator (TANK), and rs10163409 in *FTO* were among the top associations (*p* < 10^−5^) for percentage of total caloric intake from protein and carbohydrate [665]. Although the effect of FTO-mediated m^6^A-demethylation of *FGF21* mRNA has not yet been studied, recent evidence indicates that *FGF21* expression is regulated in an m^6^A-IGF2BP1-dependent manner [666]. Importantly, *FGF21* transcription is positively regulated by the Wnt pathway effector β-catenin/TCF [667,668,669]. BF with higher Wnt signaling may thus increase the expression of *FGF21* as compared to FF.

In addition to *FGF21*, breast milk lipid signaling supports the development of thermogenic AT [670,671]. MiRs are critically involved in WAT development and the browning process of adipocytes [672,673]. For instance, eicosapentaenoic acid, which is enriched together with docosahexaenoic acid in colostrum compared to mature milk [674], potentiates brown thermogenesis through binding to the free fatty acid receptor 4 (FFAR4, also known as GPR120), leading to upregulation of miR-30b-5p and miR-378 [675]. FFAR4 is expressed on adipocytes and regulates adipogenesis and energy metabolism of AT [676]. Notably, adipogenesis-related genes, including PPARγ and FABP4, are decreased in adipocytes induced from FFAR4^−/−^ MEF cells [677]. MiR-30b/c-5p targets the 3′-untranslated region of the receptor-interacting protein 140 (RIP140) encoded on *NRIP1* [161], which is a transcriptional co-suppressor of *UCP1* [678]. In contrast, miR-433 is a negative regulator of thermogenesis [679]. Thus, milk lipid signaling closely interacts with pathways of adipocyte miR signaling, which apparently interacts with a complex regulatory network of colostrum and HBME miRs.

FF, associated with an increased risk of protein overnutrition combined with the deficiency of early lactation-derived anti-adipogenic and BAT-promoting miRs, may thus result in overstimulation of ASCs in the WAT, promoting adipocyte hyperplasia. Spalding et al. [638] demonstrated that the adipocyte number is a major determinant of the fat mass in adults, and their number is set during childhood and adolescence. Future research should thus clarify the postnatal impact of FF on the number of ASCs and adipocytes during the vulnerable postnatal window of adipogenesis, influencing later metabolic health in adulthood [8,21].

Intriguingly, early lactation-derived abundant exosomal miRs (miR-22-3p, miR-1290, miR-149-3p, miR-155-5p, miR-30-5p) all target *FTO* (Table 2). Newborn infants with *FTO* gain-of-function SNPs like the rs9939609 risk allele, which is associated with increased *FTO* mRNA levels [94,95], appear to be at increased risk for postnatal aberrations of ASC homeostasis and obesity risk when receiving FF [153,154]. Excessive protein intake may further enhance already increased *FTO* expression and, combined with a deficiency of miRs in formula physiologically silencing *FTO*, may aggravate total *FTO* overexpression and FTO-mediated epigenetic dysregulation [130]. As shown in Table 3, *FTO* affects the regulation of multiple genes, including interacting cellular players (endothelial cells, pericytes, macrophages, T cells, fibroblasts, and ECM) of the SVF, all involved in the regulation of ASC homeostasis. There is a close interaction between *FTO* and Wnt signaling (Figure 3). As shown in porcine intramuscular preadipocytes, *FTO* promotes adipogenesis through inhibition of the Wnt/β-catenin signaling pathway [283].

Further evidence underscores the involvement of milk exosomes in the regulation of stem cell Wnt signaling. Treatment of intestinal epithelial cells (IEC-18) with rat milk exosomes stimulated intestinal stem cell (ISC) activity, viability, and proliferation [680]. Dong et al. [681] showed that HMDE enhanced the viability and proliferation of isolated rat neonatal ISCs due to increased Wnt/β-catenin signaling. In contrast, a specific Wnt/β-catenin signaling inhibitor (carnosic acid) decreased ISC viability significantly [681]. Following exosome administration, enhanced ISC proliferation corresponded to a significant increase in gene expression of leucine-rich repeat-containing G-protein-coupled receptor 5 (LGR5) (6.33 ± 3.01, *p* < 0.05), which was not observed in ISCs treated with HMDE-free milk (2.07 ± 0.99), compared to control (1.00 ± 0.85) [681]. LGR5^+^ stem cells have been identified at the bottoms of small-intestinal crypts [682]. LGR5, a marker for ISCs, has been identified in stem cells of actively self-renewing tissues [683]. Notably, LGR5 is a cognate receptor for R-spondins, which, together with Wnt proteins, potentiate canonical Wnt/β-catenin signaling [684]. In particular, LGR5 interacts and cointernalizes with Wnt receptors to enhance Wnt/β-catenin signaling [685,686,687,688]. R-spondins and LGRs have been found to play an important role in bone development [689]. Thus, HBME-activated LGR5-Wnt signaling is deficient in stem cells of infants receiving exosome-free formula, an unfavorable risk constellation of FF.

Both cesarean delivery and FF affect the diversity and colonization pattern of the gut microbiota during the first year of an infant’s life, increasing the risk of obesity [690,691,692,693]. Recent evidence suggests that the gut microbiota plays a role in regulating stem cell niches during early postnatal development [694], controlling neonatal Wnt signaling [695]. Human milk is a source of *Lactobacilli*, promoting their growth [696,697]. *Lactobacilli*-released lactate is crucial for activating the expression of Wnt pathway-related genes in the ISC-niche [698,699]. Specifically, lactate activates Wnt/β-catenin signaling in a G-protein coupled receptor 81 (GPR81)-dependent manner to enhance ISC-mediated epithelial proliferation [699]. Adipocytes, particularly brown adipocytes, express the lactate receptor GPR81 [700,701]. The expression of GPR81 is upregulated by PPARγ and increases during differentiation of 3T3-L1 adipocytes [702]. Importantly, a study by Verd et al. [703] showed that blood lactate levels are significantly higher (*p* < 0.5) in BF infants compared to FF infants. Furthermore, at 1 and 3 months, lactate concentrations in breast milk of term infants ranged from 137.4 to 154.8 μM. At 3 months, significantly higher fecal concentrations of lactate have been detected in BF versus FF infants [704]. Intriguingly, lactate has been identified as a critical factor for maintaining the pool of ASCs in the SVF niches [705]. In addition to lactate, FF alters the fecal concentration of short-chain fatty acids (SCFAs) during the first year of life [706]. Gut-microbiota-derived butyrate also affects Wnt signaling [707] and stimulates bone formation through T regulatory cell-mediated regulation of *WNT10B* expression [708]. Taken together, BF of infants may provide microbial metabolites that impact ASC *WNT* gene expression in the SVF, while FF infants may alter microbiome-derived metabolites that disrupt physiological Wnt signaling, increasing the risk of obesity development. The inhibitory effects of FF-induced protein overload on Wnt signaling coincide with microbiota-related changes in Wnt regulation.

The highest risk of FF is the uncontrolled preparation of infant bottles by excessive addition of formula powder, a technical problem avoided by BF. Especially, protein-rich night bottles, which extend the sleeping period of the baby, deteriorate the infant’s epigenetic programming.

The epigenetic impact of early protein overfeeding finds astonishing parallels in body size and fat body development of the honeybee queen (*Apis mellifera*) compared to the worker bees. Honeybee larvae selected for queen development receive abundant major royal jelly proteins (MRJPs), which have important functions for the developing honeybee larvae [709]. Major royal jelly protein 1 (MRJP1), the most dominant among all MRJPs, shows a high content of essential amino acids (48%) [710]. Furthermore, royalactin, a 57-kDa protein in royal jelly mimicking the effects of epidermal growth factor (EGF), induces the differentiation of honeybee larvae into queens, increasing body size by activation of S6K [711]. Activation of the PI3K/TOR/S6K pathway downstream of EGFR in the fat body stimulates insulin- and IGF-1-dependent growth in other tissues [712]. Notably, the mTOR inhibitor rapamycin, which reduces S6K activation, induces the development of worker characteristics after addition to queen-destined larvae. In accordance, knockdown of the *Apis mellifera* TOR-encoding gene *amTOR* blocks queen fate and results in individuals with worker morphology [713]. Intriguingly, Wang et al. [714] recently showed that worker larvae contain more hypermethylated m^6^A peaks than queen larvae, and many caste-differentiation-related transcripts are differentially methylated. Chemical suppression of m^6^A methylation in worker larvae reduces overall m^6^A methylation levels and triggers worker larvae to develop queen caste features. Thus, the m^6^A status functionally impacts caste differentiation and larval development and can be differentially altered by nutritional input [714]. Furthermore, the queen–worker phenotypic dimorphism is regulated by multiple histone modifications including H3K27ac, H3K4me1, and H3K36 [715,716,717]. Thus, comparative biology points to epigenetic parallels in protein and m^6^A-, as well as histone-regulated phenotype expression patterns between worker–queen dimorphism, as well as BF versus FF human infants.

The early protein hypothesis, in conjunction with an early milk miR deficiency, two adverse features characterizing FF, exert pathological epigenetic effects on ASCs, including mTORC1-S6K1-EZH2-H3K27me3-mediated Wnt suppression [258,259] combined with the deficiency of *FTO* silencing via milk miRs [130], epigenetic deviations that should be avoided by natural BF, the gold standard of postnatal epigenetic programming assured by mammalian evolution. Obviously, it was a severe mistake to regard human milk as “just food” during the early times of formula development in the last century [718]. A better understanding of the differences between epigenetic programming between natural BF and artificial FF on ASC homeostasis and development may pave the way to the prevention of our ongoing obesity pandemic.

BF provides not only nutrients but crucial signaling biomolecules required for adequate epigenetic programming of the infant [9,11,12,117,179,719]. In addition, HBM is a rich source of multipotent MSCs, which might contribute to the proper differentiation into adipogenic, chondrogenic, and osteogenic lineages under the influence of specific postpartum differentiation conditions [720,721,722]. HBM is a designated and highly sophisticated transmitter and not a simple “nutrient”, allowing appropriate postnatal programming of the infant’s ASC compartments. The transfer of HBM allows appropriate postnatal epigenetic programming, which, after sufficient application time, may make later corrective efforts of epigenetic re-programming needless [723].

Maternal obesity-associated prenatal dysregulation of epigenetic programming promoting ASC fate determination [547] may further predispose and aggravate postnatal ASC expansion induced by FF. Accumulated evidence from animal studies shows that increased fat mass and adipogenesis in offspring of obese mothers occurs early in prenatal life, indicating that the prenatal period is a vulnerable window that plays an important role in obesigenic adipocyte development in fetal AT of offspring from obese versus lean mothers [724,725,726,727,728,729,730,731,732,733,734]. Bellalta et al. [735] showed that suppressed Wnt signaling of fetal MSCs plays a critical role in the aberrant early programming of AT in offspring of women with obesity. In fact, Keleher et al. [736] recently demonstrated that MSCs from infants born to obese mothers exhibit adipocyte hypertrophy and perturbations in genes regulating adipogenesis compared to MSCs from infants of mothers with normal weight. Importantly, Boyle et al. [737] found that umbilical cord-derived MSCs from mothers with obesity were predisposed to differentiate towards adipogenic tissue, rather than myogenic tissue, due to downregulated Wnt/β-catenin signaling. Consistent with this, umbilical cord MSCs from neonates of obese mothers show a lower potential for osteogenic differentiation but an increased potential for adipocyte differentiation in vitro [738]. Human infant alterations in adipocyte- or myocyte-related cellular metabolic pathways correspond with increased adiposity and lower fat-free mass in early infancy, pointing to an adverse maternal metabolic environment that affects the fetal metabolome and epigenome, suggesting programmed differences in infant stem cell metabolism that enhance the risk of obesity [739]. In sheep, maternal obesity during pregnancy suppresses fetal myogenesis through attenuation of Wnt/β-catenin signaling [740]. Du et al. [741] have already linked reduced commitment of MSCs in fetal muscle to downregulated Wnt signaling, which attenuates myogenesis.

Therefore, maternal-obesity-associated suppression of Wnt signaling in fetal MSCs promotes ASC commitment and early steps of adipogenesis during the prenatal period, leading to aberrant metabolic programming that is postnatally continued by FF. FF of infants of obese mothers exacerbates obesigenic programming of MSCs, extending the period of dysfunctional obesigenc programming throughout the perinatal period.

Worldwide, there is an increasing prevalence of cesarean section [742], which has a negative impact on the development of a healthy intestinal microbiome [743]. Cesarean section is known to reduce the rate and duration of BF [744,745], making it a further obesigenic perinatal intervention often followed by FF, which augments adverse obesigenic programming. In conclusion, infants of obese mothers delivered by cesarean section and raised with FF are exposed to the most adverse obesigenic perinatal environment (Figure 7).

Canonical and non-canonical Wnt signaling play a crucial role in regulating stem cell proliferation and differentiation, ultimately controlling numerous developmental processes [746]. Wnt signaling is also crucial for the differentiation of pancreatic β-cells [747] and is activated during ASC differentiation into islet β-cells [748]. Physiological protein intake, along with balanced β-cell Wnt signaling may thus explain the diabetes-preventive effect of BF [20].

Future research should focus on delineating the impact of excessive protein overfeeding on Wnt-regulated adipocyte and islet stem cell homeostasis, as BF protects against the development of both obesity and type 2 diabetes [20,21].

## 4. Conclusions

Our review provides substantial evidence demonstrating that FF has an early obesogenic impact on the ASC compartment in WAT, disrupting normal postnatal Wnt signaling and promoting the development of ASCs, thus setting the stage for obesity later in life. Maternal obesity during pregnancy and a disruption of the gut microbiome by cesarean section followed by FF, are additional important perinatal risk factors that further potentiate deviations in Wnt signaling, promoting ASC expansion and increasing the risk of obesity. Infants born of obese mothers and those delivered by cesarean section may be at increased obesogenic risk due to FF, which leads to continuously uncontrolled Wnt signaling as a result of excessive postnatal protein exposure. Infants at risk for obesity due to maternal obesity or cesarean delivery should be breastfed. When BF is not an option, FF should be strictly and closely monitored by perinatal health professionals.

This man-made pandemic of obesity is a result of artificial influences on ASC priming. Obviously, BF provides optimized epigenetic control of postnatal Wnt, *FTO*, and miR signaling under the survey of the human lactation genome for a well-balanced ASC development of the infant’s WAT and BAT. BF offers nature’s most efficient, and over-millions-of-years-evolved, epigenetic strategy for appropriate AT development [18,749]. Unfortunately, BF is a still wasted opportunity for obesity prevention and global health [750]. From a medical and ethical perspective, all newborn babies should thrive on the human physiological epigenetic axis optimized and maintained by natural BF. The newborn is nursed by the mother’s breast milk, which provides the right nursing conditions for appropriate ASC development.

It is our opinion that the maternal donation of breast milk to the baby is not simply “breast-feeding” but lactation genome-surveyed correct “species-specific epigenetic programming” of the newborn, a physiological requirement and natural birthright for which artificial formula products cannot substitute.

## Figures and Tables

**Figure 1 ijms-26-04493-f001:**
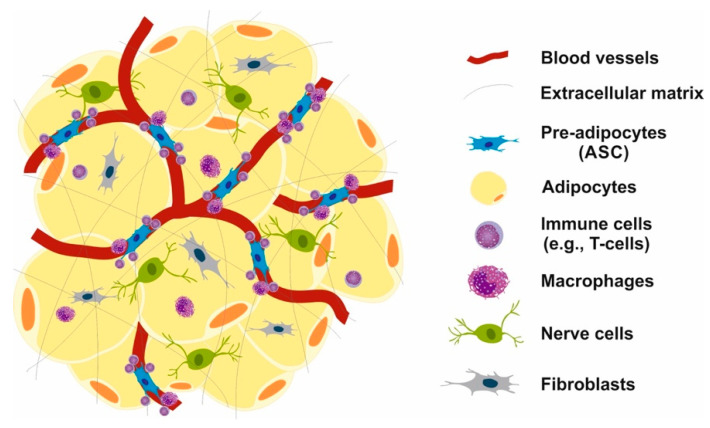
Cellular crosstalk in the vascular niche. Adipocyte stem cells (ASCs), also designated as pre-adipocytes, maintain an intimate interaction with blood vessels (endothelial cells, pericytes, mural cells), described as “blood brothers”. It is thus conceivable that ASCs are exposed to circulatory factors like hormones (insulin, IGF-1), essential fatty acids, short-chain fatty acids, lactate, and exosomes circulating in the blood stream. Other blood-derived cells like monocyte-macrophages and T cells after migration into the vascular niche may have an impact on ASC fate commitment and differentiation into mature adipocytes. Fibroblasts, extracellular matrix, and nerve cells also play a role in the cellular crosstalk that regulates adipose tissue homeostasis.

**Figure 2 ijms-26-04493-f002:**
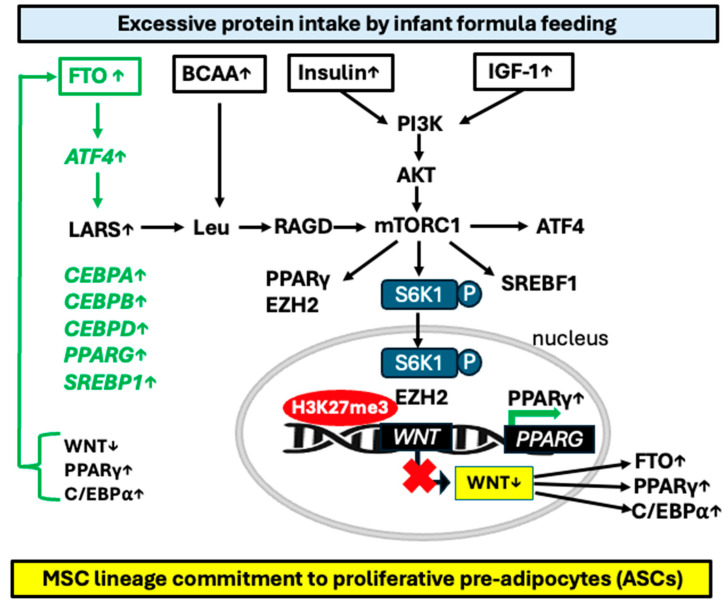
Epigenetic mechanisms linking excessive postnatal protein intake to enhanced mTORC1-S6K1 signaling resulting in *WNT* suppression and *FTO* overexpression. Insulin and IGF-1 stimulate phosphatidylinositol-3 kinase (PI3K), which activates the kinase AKT and mechanistic target of rapamycin complex 1 (mTORC1). mTORC1 phosphorylated S6 kinase 1 (S6K1-P) translocates into the nucleus promoting the recruitment of enhancer of zeste homolog 2 (*EZH2*) trimethylating histone H3 (H3K27me3), which suppresses *WNT* genes. WNT suppression enhances the expression of *FTO*, PPARγ and C/EBPα, which are crucial in determining the commitment of mesenchymal stem cells (MSCs) determining adipocyte stem cell (ASC) commitment and adipogenesis. Increased *FTO* expression, through the upregulation of activating transcription factor 4 (*ATF4*), enhances leucyl-tRNA synthase (LARS) expression. LARS, in turn, interacts with RAS-related GTP-binding protein D (RAGD) to activate leucine (Leu) signaling towards mTORC1. Consequently, excessive postnatal protein intake, leading to elevated serum levels of insulin, IGF-1, and branched-chain amino acids (BCAAs), heightens mTORC1-S6K1 activity, resulting in *WNT* suppression, thereby increasing ASC commitment. WNT suppression leads to the upregulation of PPARγ, C/EBPα, and *FTO*. *FTO*, through m^6^A RNA demethylation (depicted in green), further boosts the expression of adipogenic transcription factors such as *ATF4*, *CEBPA*, *CEBPB*, *CEBPD*, *PPARG*, and *SREBF1*. In summary, high postnatal protein intake induces significant epigenetic changes that alter MSC fate decisions towards the commitment to the adipocyte lineage, thereby increasing the pool of proliferating preadipocytes (ASC expansion). The relationship between excessive protein intake, *FTO* overexpression, and WNT suppression is closely intertwined.

**Figure 3 ijms-26-04493-f003:**
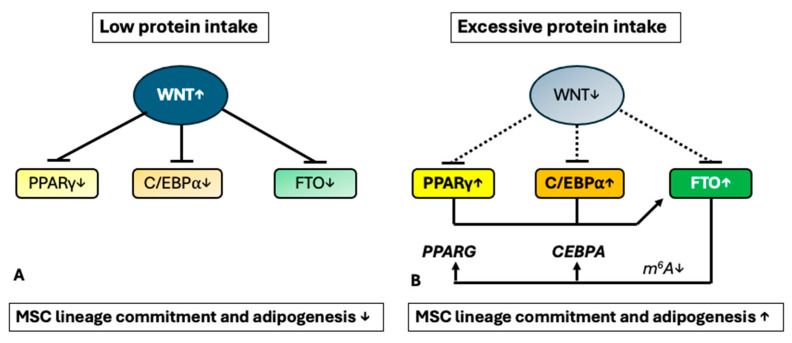
Interactive transcriptional network of the adipogenic transcription factors peroxisome proliferator-activated receptor-γ (PPARγ), CCAAT/enhancer-binding protein α (C/EBPα), and *FTO*
α-ketoglutarate-dependent dioxygenase (*FTO*). (**A**) Under conditions of low protein intake, high wingless (WNT) signaling suppresses the expression of PPARγ, C/EBPα, and *FTO*. (**B**) On the other hand, under excessive protein intake, low WNT expression enhances the expression of PPARγ, C/EBPα, and *FTO*. *FTO* further enhances the expression of *PPARG* and *CEBPA* mRNAs via m^6^A-mediated demethylation. In conclusion, Wnt signaling controls key regulators that modify MSC commitment to the adipocyte lineage.

**Figure 4 ijms-26-04493-f004:**
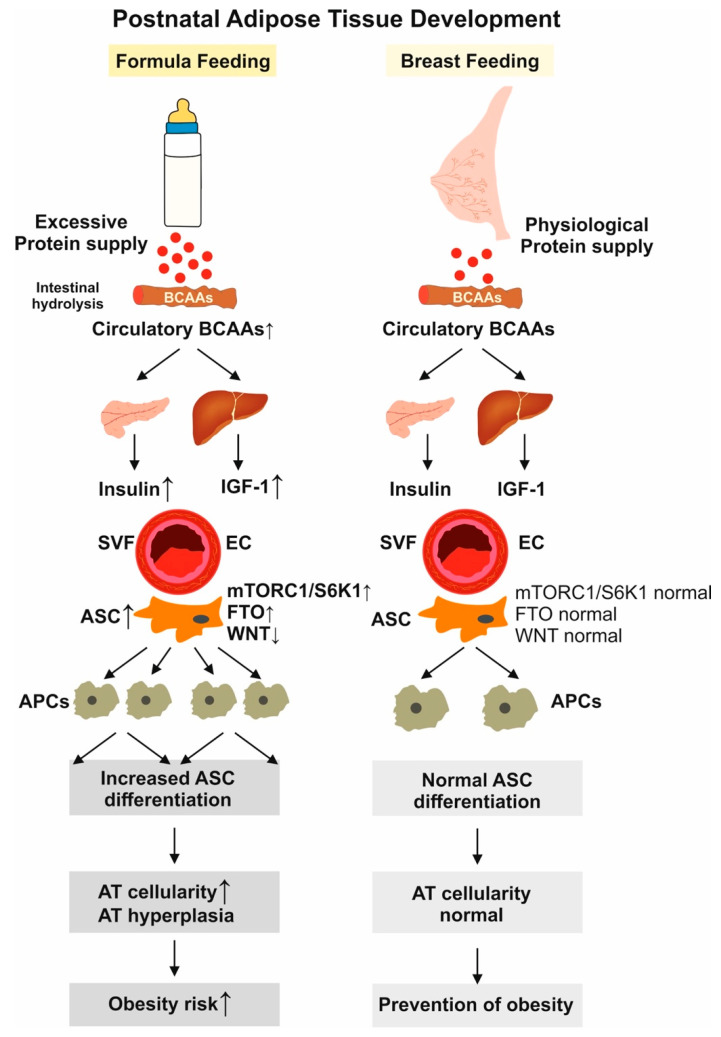
Differences in adipocyte stem cell (ASC) and adipocyte progenitor cell (APC) determination and proliferation in relation to postnatal protein intake. In formula feeding, there is increased access to branched-chain amino acids (BCAAs), which enhance serum levels of insulin, insulin-like growth factor 1 (IGF-1), and BCAAs. These compounds stimulate mechanistic target of rapamycin complex 1 (mTORC1)/S6 kinase 1 (S6K1) signaling in the stromal vascular fraction (SVF). The result is increased expression of the fat mass- and obesity-related gene (*FTO*) and reduced expression of wingless (WNT) signaling. This enhances ASC commitment and the proliferation of adipocyte progenitor cells (APCs), leading to increased adipose tissue (AT) cellularity and hyperplasia, ultimately increasing the risk of obesity. On the other hand, breastfeeding provides physiologically controlled lower amounts of protein and BCAAs compared to formula feeding. This results in significantly lower levels of serum insulin, IGF-1, and BCAAs, leading to normal mTORC1-S6K1 signaling. Breastfeeding also leads to attenuated expression of *FTO* but higher WNT signaling. These conditions allow for a physiological ASC commitment with lower APC proliferation and AT cellularity. This advantage is critical in preventing the development of obesity.

**Figure 5 ijms-26-04493-f005:**
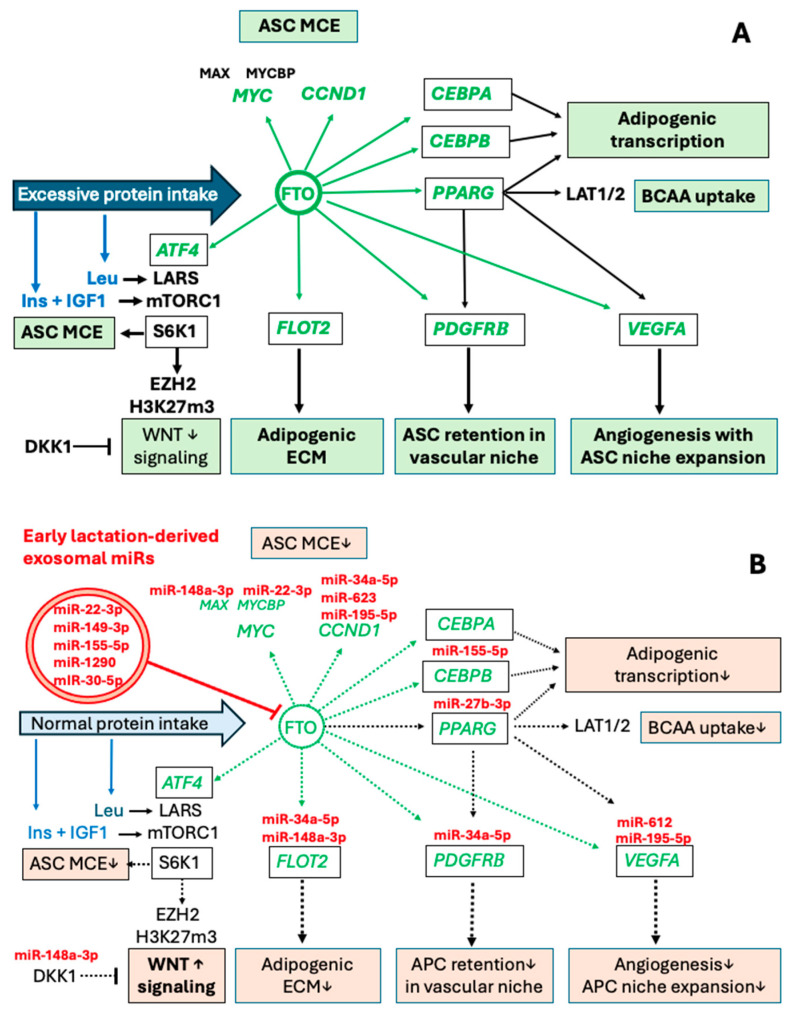
Predicted differences in microRNA (miR)-mediated adipogenic gene regulation between formula feeding and breastfeeding. (**A**) Formula feeding with increased protein intake and deficient miR supply increases the expression of *FTO*. FTO-mediated m^6^A demethylation enhances gene expression of activators of cell cycle progression (*MYC*, *CCND1*), adipogenic transcription factors (*PPARG*, *CEBPA*, *CEBPB*), *ATF4* (increasing leucine (Leu)-mediated mTORC1/S6K1 activation), upregulation of *FLOT2* (promoting adipogenic extracellular matrix (ECM) changes), upregulation of *PDGFRB* (increasing adipocyte precursor cell (APC) retention in the vascular niche as well as enhanced *VEGFA* expression (increasing angiogenesis and APC niche expansion) synergistically promoting adipocyte stem cell (ASC) mitotic clonal expansion (MCE), wingless (WNT) suppression-regulated ASC commitment and adipogenesis. (**B**) Early lactation-derived breast milk exosomal miRs attenuate *FTO* expression resulting in reduced expression of FTO-activated adipogenic transcription factors and regulators of ASC development (green color) in the vascular niche and ECM. Increased Wnt signaling attenuates ASC commitment and APC expansion adjusting ASC numbers and differentiation. Complete names of gene symbols are provided in the list of abbreviations. FTO-regulated genes are presented in green color.

**Figure 6 ijms-26-04493-f006:**
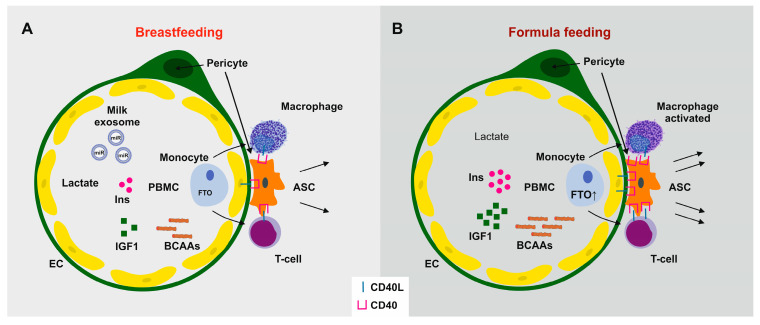
Differences in molecular and cellular crosstalk in the vascular adipocyte stem cell niche of infants receiving breastfeeding versus formula feeding. (**A**) Breastfeeding: Lower intravascular levels of insulin, insulin-like growth factor 1 (IGF1), and branched-chain amino acids (BCAAs) result in lower magnitudes of mTORC1-S6K1 signaling, leading to higher Wnt signaling that attenuates adipocyte stem cell (ASC) determination and proliferation. Milk exosomal microRNAs (miRs) may further downregulate the expression of the fat mass- and obesity-related gene (*FTO*) and adipogenic transcription factors but enhance wingless (Wnt) signaling of ASCs. Circulatory monocytes and T lymphocytes with lower *FTO* expression entering the vascular stem cell niche may exert fewer stimulatory effects mediated by CD40 ligand (CD40L) on ASC determination and proliferation. Lactate blood concentrations are also higher in breastfed compared to formula-fed infants, an additional factor potentially affecting ASC homeostasis. (**B**) Formula feeding results in increased circulatory concentrations of insulin, IGF-1, BCAAs, and increased expression of *FTO* in circulatory mononuclear cells. The absence of milk-derived exosomal miRs may enhance macrophage polarization and CD40L expression, reducing Wnt signaling that overstimulates ASC determination and proliferation, thus expanding the pool of ASCs and resulting in higher adipocyte cellularity.

**Figure 7 ijms-26-04493-f007:**
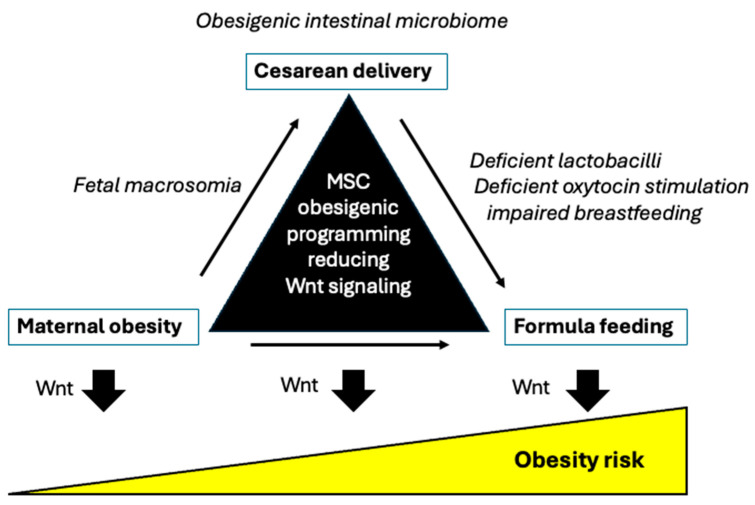
A perinatal obesigenic exposome suppresses Wnt signaling and promotes mesenchymal stem cell (MSC) commitment to the adipocyte lineage, enhancing the risk of obesity. Maternal obesity, aberrant metabolite formation of the intestinal microbiome by cesarean section, and formula feeding converge with the suppressive effects of protein-enriched formula administration on Wnt-dependent adipocyte stem cell regulation promoting obesity.

**Table 1 ijms-26-04493-t001:** Inhibitory targets of adipocyte progenitor cell differentiation and adipogenesis.

Target	Anti-ASC/Anti-Adipogenic Effect	Ref.
Deficiency of amino acids	MEFs: *FTO*↓ pS6K1↓	[109]
*FTO* knock out	MEFs pS6K1↓	[111]
*FTO* knockdown by siRNA	3T3-L1 preadipocyte differentiation↓ human ASCs: CEBPB↓ CEPBD↓ ASC differentiation↓	[130]
*FTO* targeted inhibition by miR-149-3p	Switch of BMSCs from adipogenesis to osteogenesis	[176]
*FTO* inhibition by metformin	*FTO*↓ CCND1↓CDK2↓ MCE↓ adipogenesis↓	[129]
*FTO* inhibition by ZFP217 knockout	*FTO*↓ 3T3-L1 preadipocyte differentiation↓	[187]
	MCE↓	[188]
mTORC1 inhibition by metformin	mTORC1↓ pS6K1↓ ASC stemness↑ ASC differentiation↓	[214]
mTORC1 inhibition by rapamycin	mTORC1↓ pS6K1↓	[214]
	MCE↓ C/EBP*α*↓ 3T3-L1 preadipocyte differentiation↓	[218]
	PPARγ↓ insulin action↓ adipogenesis↓	[219]
mTORC1 inhibition by raptor knockout	WAT↓ BAT↑ UCP↑ energy expenditure↑	[217]
mTORC1 inhibition by fisetin	mTORC1↓ pS6K1↓ 3T3-L1 preadipocyte differentiation↓	[266]
S6K1 knock out	MCE↓ early adipocyte progenitors↓	[218]
S6K1 inhibition LY2584702 tosylate	hepatic fat mass (steatosis hepatis)↓	[268]
S6K1 inhibition by eudesmin	S6K1↓ S6K1 nuclear translocation↓ WNTs↑ adipogenesis↓	[272]
*EZH2* inhibition by GSK126	Differentiation of MEFs into white adipocytes↓	[256]
WNT10B inhibition by WNT10B-antisera	Promotion of 3T3-L1 preadipocyte differentiation	[251]

**Table 2 ijms-26-04493-t002:** Colostrum- and milk-derived exosomal microRNAs (miRs) declining during the course of lactation and their potential inhibitory effects on adipocyte stem cell (ASC) commitment, proliferation, and adipocyte niche expansion in white adipose tissue (WAT).

MiRs	Targets	Functions	Refs
MiR-623	*CCND1* *CCND3*	Cyclin D1↓, cell cycle inhibition, MCE↓ Cyclin D3↓, cell cycle inhibition, MCE↓	[448] [449]
MiR-22-3p	*FTO* *HDAC6* *KLF6* *MAX* *MYCBP* *TIAM1* *SFRP2*	*FTO*↓, adipogenic transcription↓, MCE↓ Adipogenic differentiation↓, osteogenic differentiation↑ Fibro/adipogenic progenitors↓ Suppression of MYC signaling, MCE↓ Suppression of MYC signaling, MCE↓ T cell lymphoma invasion and metastasis 1↓, adipogenesis↓ Secreted frizzled-related protein 2↓; Wnt signaling ↑	[168,230] [459] [461] [455] [456,457] [466,467] [469]
MiR-1290	*FTO*	*FTO*↓ and inhibition of other pro-adipogenic target genes *IGF1*, *IGF1R*, *INSR*, *IRS1*, *IRS2*, *PIK3CA*, *AKT3*, *BMP4*	[472]
MiR-146a-5p	*GDF5* *TRAF6* *NEDD4L*	Inhibition of adipogenesis via GDF5-PPARγ suppression Increased degradation of AKT Wnt signaling↑, adipogenesis↓	[474] [475] [478]
MiR-195-5p	*IHH* *VEGFA* *CCND1*	Osteogenic differentiation of adipose-derived MSCs Suppression of angiogenesis Cyclin D1↓, cell cycle inhibition	[484] [485] [487]
MiR-27b-3p	*PPARG* *LPL* *SFRP1*	PPARγ↓, ASC development↓, adipogenesis↓ Lipoprotein lipase↓ ASC adipogenic differentiation↓ Secreted frizzled-related protein 1↓, Wnt signaling↑	[489,490] [493] [494,498]
MiR-34a-5p	*CCND1* *CCNE1* *CDK2* *CDK4* *CDK6* *PDGFRB* *CTRP9* *FLOT2*	Cyclin D1↓, inhibition of ASC differentiation, MCE↓ Cyclin D1↓ Cyclin-dependent kinase 2↓ Cyclin-dependent kinase 4↓ Cyclin-dependent kinase 6 Platelet-derived growth factor receptor B↓, suppression of ASC niche formation Suppression of ASC proliferation and migration Flotillin 2↓, suppression of pro-adipogenic ECM	[502] [343,504] [506] [242,243]
MiR-612	*VEGFA* *AKT2*	Vascular endothelial growth factor A↓, angiogenesis ↓ Suppression of AKT2 signaling	[511] [512,513]
MiR-149-3p	*FTO*	Switch of BMSCs from adipogenesis to osteogenesis	[175,176]
MiR-148a-3p	*ZNF217* *FLOT2* *DKK1* *KDM6B* *KDM6B*	*FTO*↓ 3T3-L1 preadipocyte differentiation↓, MCE↓ Suppression of pro-adipogenic ECM Dickkopf 1↓, osteogenesis via increased Wnt signaling Promotion of adipogenesis Suppression of MYC signaling, MCE↓	[186,187] [242,243] [514,515] [519] [520]
MiR-155-5p	*FTO* *SOX9*	Suppression of adipogenic *FTO* and lipogenesis Suppression of SOX9, C/EBPβ, PPARγ, SREBF1, FASN Reduced adipogenic differentiation of -SOX9 MSCs	[147] [530] [530]

**Table 3 ijms-26-04493-t003:** Involvement of *FTO* in m^6^A-mediated modifications of adipocytes stem cells, adipocyte stem cell niche formation, and adipogenesis.

Gene	Function	Refs
*CD44*	Upregulation of antigen CD44, ASC biomarker controlling stem cell maintenance and proliferation	[106]
*SREBF1*	Upregulation of sterol regulatory element-binding transcription factor 1, key transcription factor of lipogenesis	[128]
*PPARG*	Upregulation of peroxisome proliferator-activated receptor-γ, key transcription factor of early and late steps of adipogenesis	[129,173,174]
*RUNX1T1-S*	Upregulation of runt-related transcription factor 1, translocated to 1 short form, a key transcription factor promoting adipogenesis	[120]
*CEBPA*	Upregulation of CCAAT/enhancer-binding protein α, key transcription factor of early adipogenesis	[129]
*CEBPB*	Upregulation of CCAAT/enhancer-binding protein β, key transcription factor of adipogenesis	[124,130]
*CEBPD*	Upregulation of CCAAT/enhancer-binding protein δ, key transcription factor of adipogenesis	[130]
*MYC*	Upregulation of MYC proto-oncogene, key regulator of mitotic clonal expansion	[131,132]
*ATF4*	Upregulation of activated transcription factor 4, key regulator of amino acid signaling and adipogenesis	[134,135]
*TSC1*	Suppression of TSC complex subunit 1, a key negative regulator of mTORC1	[138]
*CCND1*	Upregulation of cyclin D1, promoting mitotic clonal expansion	[230]
*FLOT2*	Upregulation of flotillin 2, enhancing adipogenic ECM interactions and PI3K-AKT-mTORC1 signaling	[237]
*CTNNB1*	Inhibition of Wnt/β-catenin signaling via suppression of β-catenin	[283]
*BMP4*	Upregulation of bone morphogenic protein 4, stimulating adipogenensis	[285]
*FOXJ1*	Upregulation of the master ciliary transcription factor forkhead box J1, regulating ciliogenesis	[325]
*PPARGC1B*	Upregulation of peroxisome proliferator-activated receptor-γ, coactivator 1β regulation of mitochondrial function	[96,374]
*PDGFRB*	Upregulation of platelet-derived growth factor receptor β	[435,563]
*ACTA2*	Upregulation of α-smooth muscle actin	[569,570]
*VEGFA*	Upregulation of vascular endothelial growth factor A (niche formation)	[577,578]
*LINE1*	Demethylation of LINE1 RNA opening chromatin state	[585]
*CD40L*	Upregulation of CD40 ligand	[604]
*VCAM1*	Upregulation of vascular cell adhesion molecule 1 (niche formation)	[616]
*ICAM1*	Upregulation of intercellular adhesion molecule 1 (niche formation)	[616]

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
