# Peer review of "White Adipocyte Stem Cell Expansion Through Infant Formula Feeding: New Insights into Epigenetic Programming Explaining the Early Protein Hypothesis of Obesity"

_ijms, 2025, doi:10.3390/ijms26104493_

Round 1
Reviewer 1 Report
Comments and Suggestions for Authors
It is known that the intake of milk components during infancy affects obesity and metabolism during the growth process. The authors described in this review, that the molecular link between postnatal protein overfeeding and the enhancement of adipocyte stem cell expansion in adipose tissues during postnatal development in white adipose tissues. Excess protein intake activates overstimulation of insulin, IGF-1 and essential branched-chain amino acids and stimulates excessively mTORC1 and S6K1. Combination of excessive intake formula feeding-related proteins, lack of mother’s milk-derived microRNAs fail to the ability of the regulation of expression of FTO- and Wnt pathway-related proteins. This review is very informative. I learned a lot from reading it. I have no concerns.
Author Response
Reviewer 1
Thank you for appreciating our work.
Reviewer 2 Report
Comments and Suggestions for Authors
Thank you very much for allowing me to review the manuscript titled “ijms-3561145_ White Adipocyte Stem Cell Expansion Through Infant Formula Feeding: New Insights into Epigenetic FTO and Wnt Programming Explaining the Early Protein Hypothesis of Obesity”, submitted to the “Molecular Endocrinology and Metabolism” section of the journal “IJMS”.
The aim of this review is to explore the missing molecular link between postnatal protein overfeeding—commonly referred to as the "early protein hypothesis"—and the subsequent transcriptional and epigenetic changes that accelerate adipocyte stem cell (ASC) expansion in adipose vascular niches during postnatal white adipose tissue (WAT) development.
Comments:
- The title is informative and reflective of the content presented in the review. However, the use of acronyms may hinder full comprehension for readers unfamiliar with the topic. I recommend avoiding acronyms in the title or, if used, clearly defining them.
- Regarding the abstract, the objective of the review should be clearly stated. It would also benefit from specifying the type of review conducted, the time frame covered, the databases searched, and the number of articles included. These elements are crucial for establishing the review's scope and enabling comparisons with future or related reviews.
- The graphical abstract and figures included throughout the manuscript are highly informative. Nonetheless, since acronyms are used extensively, I strongly suggest including their definitions in the figure legends for clarity.
- I recommend reviewing the keywords used for indexing the article in databases. Consider verifying their alignment with the MeSH (Medical Subject Headings) classification system to enhance the discoverability of the paper.
- The introduction effectively outlines the relevance of childhood obesity and emphasizes the importance of breastfeeding, supported by pertinent literature. It also highlights the necessity for a comprehensive review. However, further clarification is needed concerning the concepts of epigenetics, FTO gene, and Wnt signaling in the context of the early protein hypothesis of obesity.
- It is also unusual to cite a reference at the very end of the introduction to state the objective. Please review lines 83 to 88, where the hypothesis and aims are presented, and consider rephrasing to reflect academic convention more closely.
- The manuscript is well-structured, with clearly delineated sections that enhance readability and comprehension.
- The conclusion should synthesize the findings from the reviewed articles and explicitly address the objective posed at the beginning of the paper. Currently, the conclusion serves more as a summary of the entire manuscript. I recommend refining this section to ensure it presents a clear and evidence-based answer to the central question.
In summary, this is a highly interesting and original piece of work that provides valuable insights into the prevention of childhood obesity through an integrated analysis of the existing scientific literature.
Author Response
Reviewer 2
TITLE: We appreciate Reviewer 2 for the critical comments.
We agree that using acronyms (FTO, WNT) in the title may be confusing for readers who are not well-versed in our research field. As a result, we gave decided to remove these acronyms and revise the title to: “White Adipocyte Stem Cell Expansion Through Infant Formula Feeding: New Insights into Epigenetic Programming Explaining the Early Protein Hypothesis of Obesity”.
ABSTRACT: Reviewer 2 suggests that the abstract would benefit from specifying the type of review conducted, the time frame covered, the databases searched, and the number of articles included. These elements are crucial for establishing the review's scope and enabling comparisons with future or related reviews.
We agree with this suggestion and have included the requested information in the revised abstract.
INTRODUCTION:
We acknowledge the criticism from Reviewer 2 regarding the late presentation of the “early protein hypothesis” in the introduction. As a result, we have rephrased the introduction to address this issue at the beginning of the chapter.
CONCLUSION: Reviewer 2 suggested that the conclusion should synthesize the findings from the reviewed articles and directly address the objective stated at the beginning of the paper. It seems that the conclusion is more of a summary of the entire manuscript. Therefore, Reviewer 2 recommended refining this section to ensure it provides a clear and evidence-based answer to the central question.
We agree with this suggestion. We have shortened and condensed the conclusion to emphasize our key findings in relation to the objective of our manuscript.
Reviewer 3 Report
Comments and Suggestions for Authors
This is a comprehensive and informative review that attempts to synthesize a vast amount of complex information from diverse fields (signal transduction, epigenetics, miRNA biology, stem cell biology, immunology) to provide a molecular framework for the "early protein hypothesis" of obesity. The authors demonstrate impressive breadth of knowledge and integrative capacity. However, several aspects concerning the structure and logical flow could be strengthened to enhance clarity and rigor.
- The central argument linking high-protein FF to ASC expansion via the mTORC1-Wnt-FTO axis is presented somewhat disjointedly. FTO, a key player, is introduced under the mTORC1 section (2.3.2) but then elaborated upon in several subsequent sub-sections (2.3.3-2.3.6) before the crucial FTO-Wnt interplay is explicitly discussed in section 2.4. This interrupts the flow and potentially obscures the centrality of FTO.
- The core mechanistic model integrating mTORC1, S6K1, EZH2, Wnt, and FTO (as depicted in Figure 2) is formally presented relatively late (2.4.1). Its components are discussed piecemeal beforehand, requiring the reader to connect the dots.
- Sections 2.6 (vascular niche details like ETS2, PDGF, α-SMA, LINE-1) and 2.7 (immune cell interactions) contain valuable information but sometimes feel appended rather than tightly integrated into the main narrative driven by FF-induced protein overload. Their direct link to the consequences of excess protein via mTORC1/Wnt/FTO dysregulation needs clearer articulation.
- The links between FF and certain downstream effects (e.g., impaired FTO degradation via GSK3, direct impacts on cilia or LINE-1 activity in infants) appear more inferential based on known pathway functions, rather than direct experimental demonstration in the context of FF.
- The review covers numerous molecules and pathways. It is difficult for the reader to discern the proposed hierarchy or relative contribution of each factor (mTORC1, FTO, Wnt, miRs, BMPs, GSK3, etc.) to the overall phenotype.
- The manuscript would benefit significantly from a distinct "Discussion" section. This would allow for a more focused and critical evaluation of the synthesized evidence, acknowledgment of limitations and potential controversies/conflicting data, comparison with alternative hypotheses, and a clearer outline of specific future research directions needed to test the proposed model. Currently, such elements are scattered or brief.
Author Response
Reviewer 3
We would like to express our gratitude to Reviewer 3 for providing valuable, meaningful and extensive comments.
Reviewer 3’s main concern is that the major pathway driving ASC commitment, which we believe to be the mTORC1/S6K1/EZH2/WNT axis, is not sufficiently emphasized.
Therefore, we have highlighted the mTORC1/S6K1/EZH2/WNT axis in the revised abstract, discussion and conclusion. The first figure illustrating the central pathway (Figure 2) elaborates on this key signaling axis. We have provided additional evidence that activated mTORC1 enhances the protein translation of EZH2, further supporting this central pathway (Harachi et al., Cancer Res. 2020;18:1142-1152). We have included this information in the reference list and modified Figure 2 accordingly.
The impact of FTO on ASC commitment is complex and highly interactive. FTO serves two key mechanisms: 1) FTO senses essential amino acids, activating mTORC1 and thus influencing EZH2-mediated histone methylation suppressing Wnt signaling
and
2) via its m6A-demethylase activity promotes the mRNA expression of various key adipogenic transcription factors targeting MSCs, ASCs, and multiple cells of the ASC nursery niche. Therefore, there is not a single pathway driving the epigenetic mechanisms of early protein-induced obesity, but rather a complex interacting network augmenting adipogenic signaling in a converging manner. In fact, we have shown that FTO, Wnt, PPARgamma and C/EBPalpha interact with each other, as visualized in Figure 3.
To enhance the final synopsis of the central mTORC1/EZH2/Wnt pathway, its interactions with upregulated FTO and the apparent fine-tuning by milk miRNAs, we have restructured the discussion and conclusion to highlight the hierarchy and interplay of obesigenic Wnt signal transduction induced by postnatal protein overfeeding.
We hope that these changes have improved the comprehension of our findings and addressed the understandable concerns of Reviewer 3.
Additionally, we have suggested future research directions focusing on the Wnt pathway, which also plays a crucial role in pancreatic beta-cell differentiation. We have provided a new reference (Wang et al., Biomed. Res. Int. 2017;2017:2501578) at the end of the revised discussion.